# Bradyzoite subtypes rule the crossroads of *Toxoplasma* development

Arzu Ulu [1], Sandeep Srivastava[1,2], Nala Kachour[1], Brandon H. Le[3], Michael W. White [1] & Emma H. Wilson [1]

Toxoplasmosis is a major risk to chronically infected individuals, especially those who become immunocompromised. Although one-third of the globe is infected with *Toxoplasma*, no treatments prevent or eliminate cysts in part due to limited understanding of bradyzoite biology. The cyst is central to Toxoplasmosis, as transition from bradyzoites to tachyzoites drive pathology. In this study, we aim to understand the biology of bradyzoites prior to recrudescence and the developmental pathways they initiate. Here, we discover ME49EW cysts from infected mice harbor multiple bradyzoite subtypes with distinct fates. Purified subtypes exhibit defined developmental pathways in animals and in primary astrocytes. Single-bradyzoite RNA-sequencing reveals five major subtypes within cysts. We further show that a crucial subtype in chronically infected mice is absent from a widely used in vitro model of bradyzoite development. Altogether, this work establishes new foundational principles of *Toxoplasma* cyst development and reactivation that operate during the intermediate life cycle of *Toxoplasma*.

*Toxoplasma gondii* is a globally prevalent protozoan parasite, infecting nearly one-third of the human population. During chronic infection, the tissue cyst form is essential for both parasite survival and disease pathogenesis. In immunocompromised individuals, reactivation of these cysts or recrudescence, is the only known cause of acute, and sometimes fatal, toxoplasmosis[1,2]. Cyst reactivation is typically controlled by IFN-γ mediated immune responses[3]. However, with rising immunosuppression in the U.S. population, 6.6% of adults are immunocompromised, and with over a million cancer patients undergo chemotherapy annually, the risk of cyst reactivation becomes a growing public health concern[4]. Given its high prevalence[5,6], and potential for severe disease, understanding bradyzoite biology and tissue cyst dynamics is critical.

Broad host range and durable *Toxoplasma* tissue cysts underlie its success as an intracellular parasite. Transmission occurs via feline oocysts and ingestion of cyst-contaminated foods. Bradyzoites within tissue cysts are the pivotal stage of the *Toxoplasma* life cycle. They serve as a reservoir of infection capable of differentiating directly into tachyzoites responsible for dissemination and cell lysis[7–9] or switching to the merozoite stage in the feline definitive host that leads to sexual reproduction in the cat gut[10–12]. Bradyzoites are also capable of direct replication, which is thought to be required for maintaining cyst burden and chronic infection[10,11,13–15]. Investigating this slow growing, in vivo bradyzoites is difficult, but our ex vivo recrudescence system using in vivo-derived cysts[13] allows characterization of growth dynamics and heterogeneity around reactivation that cannot be fully replicated in vitro.

In the *Toxoplasma* Type II ME49 strain, genes encoding SAG1-related surface proteins (SRS) includes 144 genes; 109 that match gene models and 35 incomplete pseudogenes[16,17]. SRS genes are present on all ME49 chromosomes with clusters of SRS paralogs found on seven chromosomes. SRS antigens discovered 40 years ago remain an enigmatic class of *Toxoplasma* factors[18,19]. SRS antigens influence host cell adhesion[20], and it is postulated that immunodominant SRS antigens, such as SAG1, elicit immune responses against the tachyzoite stage so that bradyzoites, which lack SAG1 are spared[21].

[1]Division of Biomedical Sciences, School of Medicine, University of California, Riverside, Riverside, CA, USA. [2]Bharat Serums and Vaccines Limited, Navi Mumbai, Maharashtra, India. [3]Department of Botany and Plant Sciences, Institute for Integrative Genome Biology, University of California, Riverside, CA, USA. e-mail: michaewh@ucr.edu; emma.wilson@ucr.edu

Notwithstanding, the realization of purpose for a few SRS antigens, the function of most SRS antigens remains unknown. However, antibodies raised against developmentally regulated SRS antigens are valuable reagents in the study of *Toxoplasma* developmental pathways[19,22]. Similarly, SAG1 is a primary marker of the tachyzoite stage, while induction of SRS9 expression (i.e. SRS16B) indicates early bradyzoite development[13]. In this study we investigated the SRS22A protein that belongs to a cluster of 9 paralogs (SRS22A-I) and one pseudogene tandemly arrayed on the left end of chromosome VI. The *SRS22* gene family are highly expressed in merozoites of the definitive life cycle[23,24] with the exception of *SRS22A* mRNA, which is additionally expressed in bradyzoites[23].

In this study, we demonstrate SRS22A is a dominant bradyzoite surface protein expressed in ME49EW bradyzoites from mice but is not detected in conventional in vitro models of bradyzoite development such as alkaline- stress model which now includes our ex vivo bradyzoite recrudescence model[13]. We determined that ME49EW tissue cysts from infected mice harbor at least two bradyzoite subtypes distinguishable by *SRS22A* expression that have different growth and developmental trajectories. Using single bradyzoite mRNA-sequencing we confirm that in vivo bradyzoites differentially express *SRS22A* mRNA and provide evidence that there are additional bradyzoite subtypes likely present in the tissue cysts of chronically infected mice.

## Results

### Bradyzoites from chronically infected mice have unique surface antigen composition

It is generally understood that conventional alkaline-stress models of tachyzoite-to-bradyzoite differentiation do not faithfully reproduce the full breadth of native developmental processes. The reliance on in vitro methods has contributed to significant gaps in our understanding of native bradyzoite developmental biology in the *Toxoplasma* intermediate life cycle. To begin addressing this problem, we sought to identify potential in vivo bradyzoite markers by comparing the mRNA expression of *SRS* genes in ME49EW bradyzoites from 30-day infected mice against in vitro ME49B7 tachyzoite and alkaline-stressed parasite mRNA expression (FPKM_fold-change, for full RNA-seq data see ref. 13). In order to compare in vitro marker expression with the markers observed in vivo, we examined the most commonly employed in vitro system in which alkaline stress is used to generate in vitro ME49B7 bradyzoites. Seventeen SRS antigen mRNAs were expressed at least 10-fold higher in ME49EW bradyzoites from mice (Fig. 1a table). The mRNA encoding SRS9 (also designated as SRS16B) was highly expressed in ME49EW bradyzoites as well as alkaline-stressed in vitro bradyzoites validating the use of anti-SRS9 antibodies as an early bradyzoite marker. The gene encoding the SAG-related protein SRS22A was the top induced SRS mRNA in ME49EW bradyzoites from in vivo cysts at >1000-fold higher levels than *SRS22A* mRNA levels in tachyzoites. The in vivo expression of *SRS22A* mRNA was also well above SRS22A mRNA levels from alkaline-stressed ME49B7 parasites (~17 fold higher, Fig. 1a). We selected a peptide from the N-terminal region of the SRS22A protein coding sequence (*LRGNDGRSSRVIEKEAEVAK*; Fig. S1A) based on informatic prediction of immunogenicity and produced peptide-specific antibodies in rabbits. The SRS22A antisera successfully stained in vivo tissue cysts from mice (30-days post-infection), whereas there was no reactivity of the pre-immune serum control (Fig. S1B). Furthermore, Western blot analysis detected a strong, single band in excysted bradyzoites obtained from 30-day infected mice with no signal detected in RH tachyzoites grown in vitro (Fig. 1b) indicating specificity for bradyzoites[25].

Utilizing the SRS22A antiserum, we determined the kinetics and cyst distribution of SRS22A expression in tissue cysts from infected brain homogenates at 14-, 21-, 28-, and 35-days post-infection (Fig. 1c–f). As examined by others, there are likely complex patterns of cyst growth, rupture and reformation over the course of infection, however, average cyst size generally increases over this period[11–13], which is consistent with our results here (Fig. 1c). The proportion of cysts expressing SRS22A is generally correlated with increasing cyst size (Fig. 1d). By the last timepoint (Day 35), >75% of tissue cysts showed some expression of SRS22A. Over the course of infection, we estimated that cysts exhibited three SRS22A bradyzoite configurations; near 100% positive or negative or mixtures of SRS22A+/SRS22A-bradyzoites (Fig. 1f). The proportion of ~100% positive SRS22A cysts increased from Day 14 to Day 21 post-infection, and this level of expression was maintained at ~20% of total cysts thereafter (Fig. 1e). By contrast, the proportion of cysts with mixtures of SRS22A- and SRS22A+ bradyzoites continued to increase over time reaching >50% of the population by Day 35. A process of cyst maturation has been postulated to explain the differences in cysts generated by in vitro alkaline-stress models (considered to be early cysts) and the putative mature cysts recovered from >30-day infected mice. By probing for SRS22A, (an indicator of in vivo cysts) further cyst heterogeneity is revealed and indicates there is no single "mature" cyst in the native intermediate life cycle, but that cyst heterogeneity exists throughout the course of *Toxoplasma* infection in the brain.

### SRS22A is exclusively expressed by in vivo generated cysts

We next determined whether SRS22A was expressed in bradyzoites that develop in the in vitro alkaline stress-model. *SRS22A* mRNA expression in ME49EW cysts from mice is substantially higher than in cysts generated under alkaline-stress treatment (Fig. 1A). To determine if SRS22A protein expression differentiates in vivo from in vitro bradyzoites, we used ME49B7 tachyzoites[26] to infect HFF monolayers, and following a 4 h invasion period, shifted half the cultures to pH 7.4 media and the other half to pH 8.2 bradyzoite-media. We co-stained infected HFF cells with two pairs of antibodies; the tachyzoite-specific SAG1 monoclonal antibody with the bradyzoite-specific anti-SRS9 or anti-SRS22A rabbit antisera. Parasite vacuole size and SRS antigen expression was assessed at 48 h after the shift to pH 8.2 media (Fig. 2a). ME49B7 tachyzoites grown in pH 7.4-media actively replicated and expressed SAG1 (Fig. 2a). In pH 8.2-media, ME49B7 tachyzoites grew much slower as evidenced by the smaller range of vacuole sizes. Tachyzoite vacuoles (SAG1+ only) were 41% of the total parasite population whereas 59% of vacuoles co-stained for SAG1+ and SRS9+ (SAG1+/SRS9+ vacuoles) or expressed only SRS9+ antigen and considered to be bradyzoite vacuoles. No parasites in either pH 7.4- or pH 8.2-media expressed detectable SRS22A antigen.

In our ex vivo recrudescence model, in vivo ME49EW bradyzoites are used to infect primary astrocytes cultivated under low oxygen (5%) conditions[13]. As expected, SRS22A antisera stained excysted ME49EW bradyzoites from in vivo cysts as well as single bradyzoites that had invaded astrocytes (Fig. 2b, free and Day 1 bradyzoites). Most vacuoles in Day 1-infected astrocytes contain single (pre-replication) bradyzoites reflecting time needed for dormant bradyzoites to reawaken[13]. In advance of the first parasite doubling, we detected switches in SRS antigen composition indicating changes in developmental transcription (e.g. SRS22A) occur before replication commences in full. The two-representative single bradyzoite vacuoles shown here (Fig. 2b, D1 EW) both express SRS22A, while only one vacuole co-stained brightly for SAG1. In the ex vivo recrudescence model[13] replicating bradyzoite vacuoles occur alongside tachyzoite vacuoles (SAG1+) as visualized by DBA positive cyst walls (Fig. 2b); D3 EW and D7 EW image panels). Importantly, SRS22A expression was not observed in these mixed populations (Fig. 2b) in either the fast-replicating tachyzoite period at Day 3 (D3 EW) or the post-growth shift tachyzoite/bradyzoite population at Day 7 (D7 EW). Similar to the alkaline-stress model SRS22A was not detected in any parasite vacuole beyond Day 1 in the infected astrocytes.

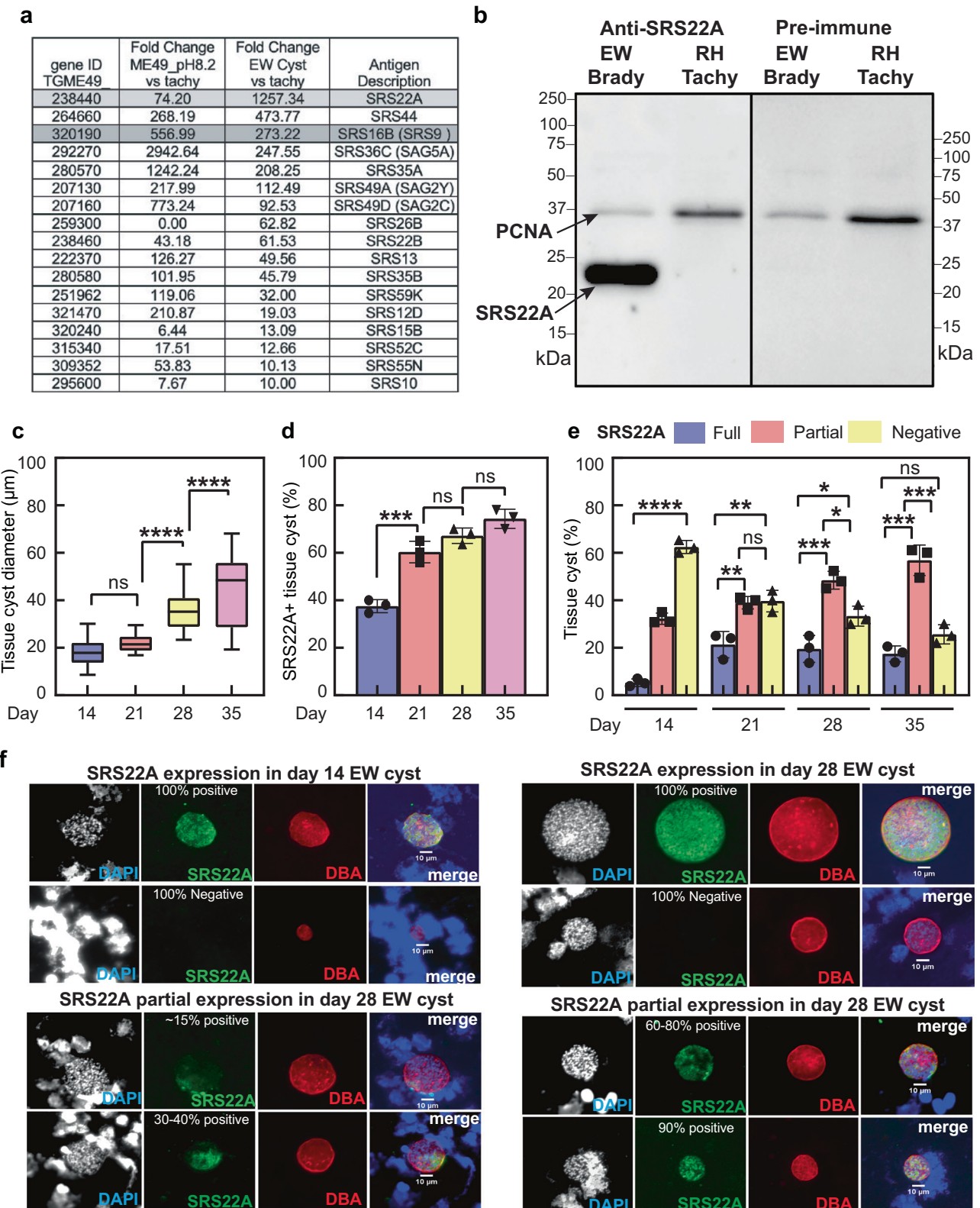

**Bradyzoite subtypes initiate different developmental pathways**
The discovery of SRS22A+ bradyzoites in tissue cysts of infected mice, but not in the alkaline stress-model of cyst development, raised questions about the developmental trajectory of both SRS22A- and SRS22A+ bradyzoite subtypes. The finding that cysts from chronically infected mice had mixed SRS22A (+ and -) bradyzoite populations (Fig. 1e) gave equal importance to investigating SRS22A- bradyzoites.

Therefore, to determine the developmental paths of these bradyzoite subtypes, we sorted SRS22A+ and SRS22A- bradyzoites from brain tissue cysts by FACS (Fig. 3, Fig. S2). Identification and sorting of parasite populations revealed a distinct SRS22A+ expressing population representing approximately 60-80% of all Day 0 bradyzoites (Fig. S2). To determine any differential SRS22A-associated growth patterns, sorted populations were used to infect primary astrocyte

**Fig. 1 | Identifying in vivo bradyzoite-specific SRS genes. a** List of 17 SRS genes with at least 10-fold higher expression in ME49EW in vivo bradyzoites (40-days post-infection) as compared to ME49B7 tachyzoites grown in vitro and ME49B7 tachyzoites under 48 h of alkaline-stress (pH 8.2). SRS22A is the top expressed SRS gene in ME49EW in vivo bradyzoites, surpassing even SRS9 (SRS16B), an early canonical bradyzoite marker. SRS22A was chosen for peptide-based antibody development (Fig. S1a). **b** Western blot using rabbit anti-SRS22A antiserum detects a strong signal for SRS22A ( ~ 22 kDa) in ME49EW bradyzoites (excysted from in vivo cysts) that is absent from RH-tachyzoites. PCNA (loading control), and a pre-immune serum-control are shown. **c–e** Quantification of ME49EW tissue cysts in brain homogenates from CBA mice infected for 14-, 21-, 28- and 35- days: **c** Tissue cysts in mouse brain increase in size over time. Boxplots are median (centre line), IR (25th-75th percentile; box), whiskers indicate the min-and-max values. One-way ANOVA with Tukey's multiple comparisons (main effect *P*-value < 0.0001 (F = 96.68)). **d** Percentage of tissue cysts harboring SRS22A+ bradyzoites (data are mean ± SD, and one-way ANOVA main effect *P*-value < 0.0001 (F = 72.15), and **e** Quantification of three SRS22A cyst phenotypes; 100% SRS22A + , partial SRS22A+ and 100% SRS22A- bradyzoites at the four timepoints. Data are mean ± SD. One-way ANOVA main effect *P*-value < 0.0001 (F = 49.86). At least two brain homogenates were analyzed per timepoint, counted 3x. ****P < 0.0001, ***P < 0.001, **P < 0.01, *P < 0.05. **f** Representative immunofluorescence images of tissue cysts at days 14- and 28-post-infection, showing SRS22A+ versus SRS22A- expression (homogenous or mixed). Scale bar = 10µm. Brain homogenates were co-stained for SRS22A (green), DBA (red), and DAPI (blue). Note that the interior regions of cysts that lack anti-SRS22A staining have numerous parasite nuclei. A decolorized DAPI image was included to better visualize parasite nuclei in cysts. At least two brain homogenates were analyzed for each time point. Source data are provided as a Source Data file for this figure.

cultures (Fig. 3a). Population growth and SRS antigen expression patterns (SAG1 versus SRS9) were determined during the maximal ex vivo growth stage at Day 5 post-infection. The infected cultures revealed clear differences in growth rates. SRS22A + _seeded cultures expanded 3.1-fold by Day 5, whereas parasite populations from SRS22A-_seeded cultures only reached replacement levels (1.1-fold). Remarkably, the end-stage parasites (tachyzoites or bradyzoites) produced by these cultures were also different. On average over 80% of sorted SRS22A expressing bradyzoites converted to replicating tachyzoites (SAG1+ only) with very few bradyzoite vacuoles (2.3%, SRS9+ only). In contrast, bradyzoites that were originally SRS22A- led to the opposite developmental profile and replicated as bradyzoites expressing SRS9 and forming DBA positive cysts (Fig. 3a, b). Here, bradyzoites (30% of vacuoles, SRS9+ only) were the principal end-stage produced with few tachyzoite vacuoles present (3.7% vacuoles SAG1+ only). Tissue cyst wall assembly was only detected in the SRS9 + /SAG1- vacuoles (Fig. 3b). We used a restrictive criterion for calling end-stages, and thus, in both SRS22A infections there were intermediate parasites (see examples Fig. S1C) co-expressing varying levels of SAG1 and SRS9 antigens (pairwise dull to bright staining; 16% in SRS22A + , >60% in SRS22A-) and made up the remaining proportion of parasites.

The preference of SRS22A- bradyzoites to initiate further brady-zoite replication suggests two major aspects of bradyzoite-cyst biology: firstly, that there is heterogeneity of bradyzoites within cysts and secondly that these subpopulations have preferential developmental trajectories. Our previous work revealed heterogeneity of bradyzoites during ex vivo recrudescence with a proportion of parasites (~20%) replicating as SRS9+ bradyzoites (brady-brady replication) in addition to the canonical brady-tachy pathway associated with cell lysis and pathology during reactivation[13]. To determine if these subpopulations of bradyzoites also represent a stable developmental phenotype, SRS9+ bradyzoites were sorted from recrudescing populations in astrocytes and evaluated for growth using SAG1 versus SRS9 antigen expression at Day 1-, 3- and 5-days post-infection (Day 7 parasites, Fig. 3C). As we demonstrated in Fig. 2, Day 7-bradyzoites sponta-neously assemble tissue cyst walls (DBA + ) but do not express SRS22A, and are therefore, SRS9 + /SRS22A-. Pre-sort, SRS9+ parasites (under-going brady-brady replication) represented ~8% of the population (Fig. 3C) and as expected sorted parasites retained the SRS9+ only pattern 24 h after inoculation (Fig. 3C, Day 1). However, these parasites continued to express only SRS9+ over the course of the next five days and did not express SAG1. It was apparent that these bradyzoites retained their ability to replicate, evidenced by the increase in SRS9+ only vacuole size (Fig. 3C, column numbers). After 5 days of sorted parasites being in culture ~75% of vacuoles remained brady-brady replicating parasites successfully making cyst wall proteins (Fig. 3D). A proportion of parasites (~20%) did convert to SAG1+ only vacuoles, however, these parasites were not growing well (average vacuole size of <2). It is notable that tachyzoites in the pre-FACS Day 7-population grew better with an average vacuole size of 14.3 likely because the parentage of these parasites was the fast-growing tachyzoite[13]. Alto-gether these data demonstrate that in vivo SRS22A- as well as in vitro SRS9 + /SRS22A- bradyzoites are primary initiators of direct bradyzoite expansion, while in vivo SRS22A+ bradyzoites are responsible for recrudescence into the fast-growing tachyzoites.

## Infection potency requires SRS22A+ bradyzoites

To evaluate whether FACS-purified SRS22A+ and SRS22A- bradyzoite populations are developmentally competent, we infected CBA mice with each subtype and assessed parasite burden during acute (5-days post-infection) and chronic infection (28-days post-infection). At the acute time point, SRS22A+ infections produced significantly more SAG1+ tachyzoites and vacuoles in peritoneal exudate cells (PECs) (Fig. 4a). Parasite burden in the liver, spleen and lung also trended higher in SRS22A+ infections (Fig. 4b). At 28-days post-infection, mice infected with SRS22A+ bradyzoites developed significantly more cysts than those infected with SRS22A- bradyzoites (Fig. 4c), consistent with the robust dissemination capacity of the fast-growing tachyzoites[13] derived from SRS22A+ bradyzoites.

## Single cell RNA-seq indicates evidence of multiple bradyzoite types

The above data using only three antigens, demonstrate heterogeneity of bradyzoites that predicts the developmental trajectory of parasites. To further understand the underlying basis for this heterogeneity, we analyzed the transcriptome of single, excysted ME49EW bradyzoites from 40-day infected mice (Figs. 5, 6, S3, and S4). Individual brady-zoites from two brain tissue cyst harvests collected more than 6 months apart (Fig. S3A, r = 0.965) were captured using the 10x Chro-mium Single Cell Gel Bead kit and 3′- RNA-seq libraries sequenced, yielding ~22 billion reads with >80% mapping to the *Toxoplasma* genome and 4,190 high-quality bradyzoites (Fig. S3B). UMAP projec-tions (R v4.4.2; Seurat 5.1.0) of bradyzoites from each cyst harvest produced matching parasite clusters (Groups A-E) (Fig. S3C). The merged data (Fig. 5a, UMAP projection) revealed ~1200–3300 mRNAs per bradyzoite Group (*P* val <0.05, Supplementary Data 1) with 374 genes expressed in all Groups (Fig. 5b, Venn diagram). Group C, which interfaces with Groups A, B, and D, had the fewest (<75) unique tran-scripts. Approximately 40% of *Toxoplasma* genes are cell-cycle regu-lated, with peak transcription in G1 or in S/M phases[27]. Prior studies on tachyzoite cell cycle uncovered that bradyzoite-specific gene expres-sion may also be linked to S/M phases[27] and our scRNA-sequencing supports this: Group A-E show nearly twice the expected (17%) pro-portion of S/M-enriched transcripts (except Group D), while G1 levels match predictions (~23%) (Fig. 5c). Nevertheless, these transcriptomes did not phenocopy the synchronous cell-cycle contours seen in ex vivo ME49EW tachyzoites[13], and principal component analysis indicates that neither cell-cycle stage nor canonical bradyzoite markers (*BAG1, LDH2, ENO1*) drive major variation (Fig. S3E), unlike alkaline-stress

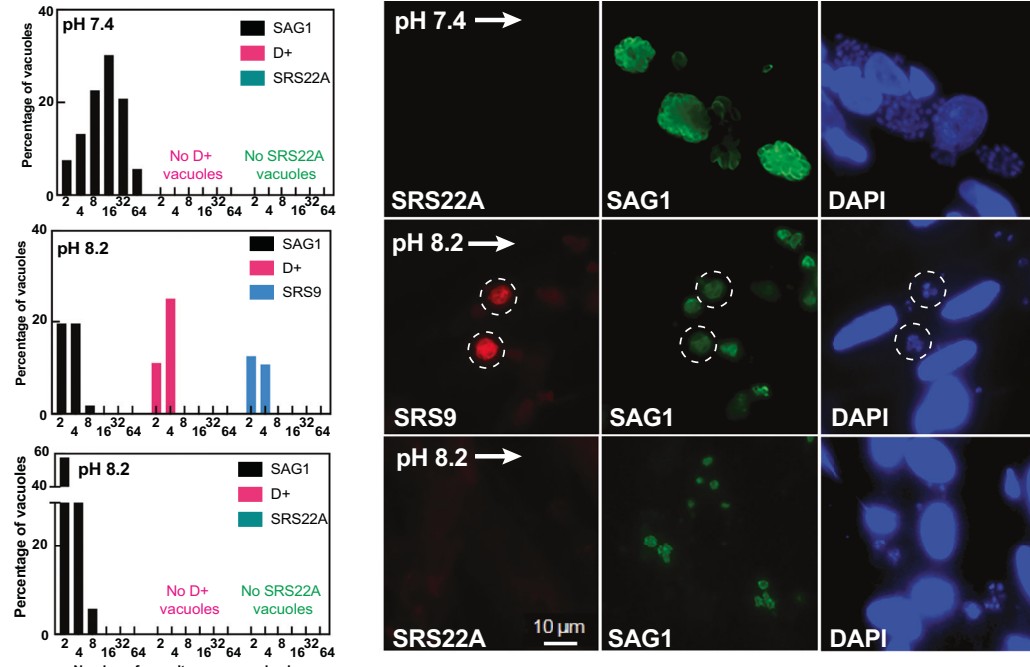

**a SRS22A is not expressed in the pH-stress model of bradyzoite development**

**b Expression of SRS22A in the ex vivo bradyzoite recrudescence model**

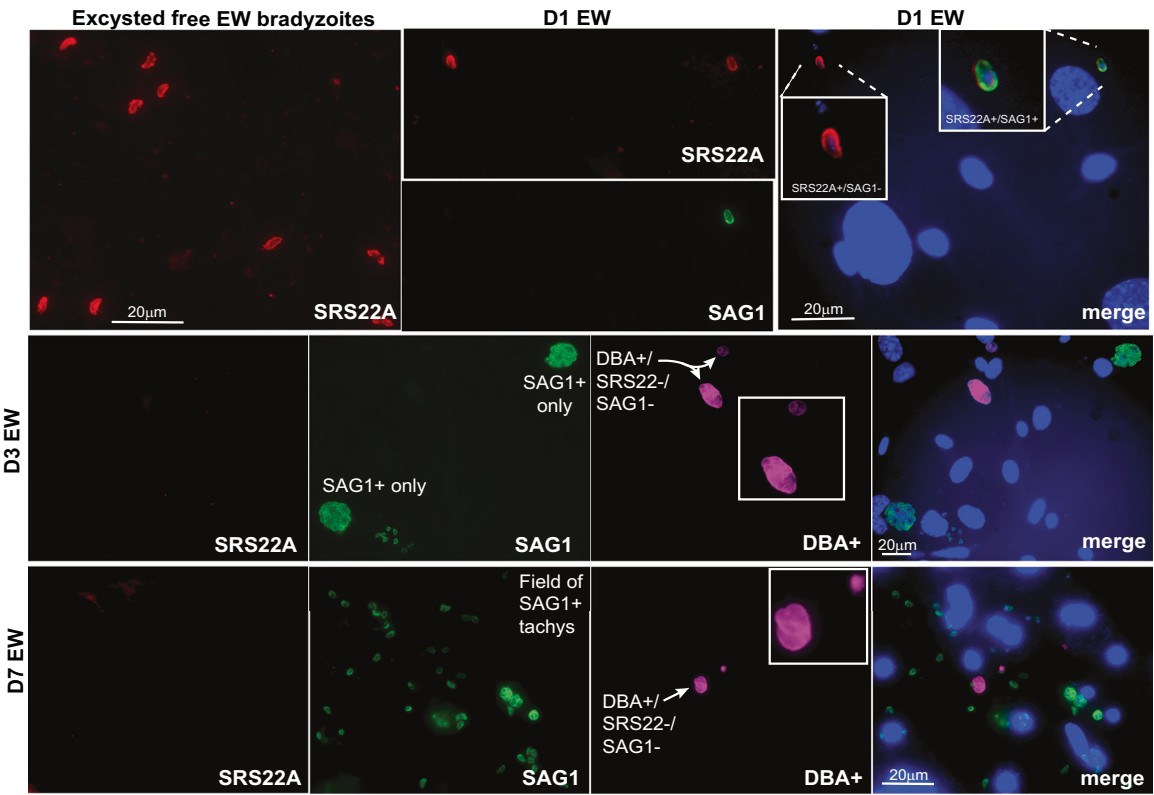

studies[28,29]. Because cell-cycle markers are indirect measures of cell proliferation, DNA content remains the most accurate metric. Our previous flow cytometric analysis showed ~2% of these bradyzoites possess a 1 N + DNA content[13]. Group E, which expresses the most S/M transcripts (745 mRNAs) and elevated growth-related mRNAs (Fig. S3D; Fig. S4C), likely represents this small replicative subset within otherwise dormant cystpopulations[13].

To determine the possible functional differences among the clusters, differentially regulated transcripts in Groups A-E were compared to the transcriptome of Day-7 BAG1+ME49EW bradyzoites. These bradyzoites spontaneously form in primary astrocytes following recrudescence[13] and represent a replicating, cyst forming and developmentally competent population of bradyzoites in which to compare our originating bradyzoite cyst clusters. A combined UMAP projection

**Fig. 2 | Cell culture models of bradyzoite development fail to express SRS22A.**
**a** (right images) SRS22A expression was not detected in any parasite vacuole irrespective of the pH condition used. The presence of SRS9+ vacuoles that were the result of tachyzoite-to-bradyzoite development in pH 8.2 media (48 h) confirm early bradyzoite differentiation in these cultures. The infected HFF cell cultures were fixed and co-stained for SRS9 (bradyzoites, red) and SAG1 (tachyzoites, green), or separately for SRS22A (red) and SAG1 (green), which was required because SRS9 and SRS22A antisera were raised in rabbits. DAPI was included to visualize nuclei. Scale bar = 10 μm. (left graphs) Quantification of vacuole size for SAG1+ only, both markers (D+ indicates vacuoles with both markers: SAG1 + / SRS9 + ), and SRS9+ only vacuoles demonstrated cultivation in pH 8.2 media caused overall slower parasite growth, which is known to be associated with the alkaline-stress bradyzoite differentiation model. Alkaline-shift experiments were repeated three times and all vacuoles in each coverslip were counted. **b** Immunofluorescence images showing SRS22A staining in ME49EW excysted, free bradyzoites and ex vivo bradyzoite-infected primary astrocyte cultures at days 1-, 3-, and 7- post-infection. Free bradyzoites and infected-astrocyte cultures were co-stained for SRS22A (red), SAG1 (tachyzoite-specific, green), and DBA (cyst wall-specific, magenta). Notably, SRS22A staining diminishes very quickly (within the 1st 24 h) in ex vivo bradyzoite-infected astrocytes, and thus, at day 3 (D3 EW) and day 7 (D7 EW) post-infection no parasites stained positive for SRS22 A. Note that SRS22A was also not expressed or re-expressed in bradyzoite vacuoles where cyst wall formation was active (DBA + ). Scale bar = 20 μm. Ex vivo experiments were repeated at least three times. Source data are provided as a Source Data file for this figure.

of the in vivo versus ex vivo bradyzoites demonstrated clear separation of the in vivo bradyzoite Groups (A-E) from the distinct cluster of D7-BAG1+ bradyzoites from astrocytes (Fig. S3F). Hundreds of up-and down-regulated mRNAs were detected in Group A-E bradyzoites (Fig. S3F) compared to D7-BAG1+ bradyzoites (log2 fold-change results, Supplementary Data 2) with many of the highest up-regulated mRNAs encoding recognized canonical bradyzoite protein markers (e.g. BAG1, LDH2, ENO1), selective rhoptry and GRA proteins, ApiAP2 factors, and SRS proteins (Fig. 5c; Supplementary Data 2). Considering the significance of these proteins in *Toxoplasma* biology, we also compared their expression among Groups A-E. *AP2* and *SRS* mRNA profiles (Supplementary Data 1) differed by bradyzoite Group, and *AP2IX-9* was the highest expressed ApiAP2 factor in Groups A-D (Fig. S4A). The mRNA encoding tachyzoite-specific antigen, *SAG1*, was the most significantly down-regulated transcript in Group A-E bradyzoites (Fig. 5d, and Fig. 6a). The presence of *SAG1* on the surface of parasites is a distinguishing marker of bradyzoites that develop in cell culture models from the in vivo bradyzoites of infected mice[13].

In the experiments above, we established the patterns of in vivo bradyzoite SRS22A antigen expression and determined the distinct developmental of SRS22A+ and SRS22A- bradyzoite subtypes (Fig. 3). Thus, the expression profile of SRS22A mRNA in bradyzoite Groups A-E was of considerable interest. Indeed, SRS22A transcript expression was found highly focused in Group B bradyzoites accounting for 81% of the bradyzoites (Fig. 6a). Consistent with the lack of SRS22A protein expression in cell culture-formed bradyzoites (Fig. 2), we did not detect *SRS22A* mRNA in D7-BAG1+ bradyzoites (Fig. S3G). Several other features of Group B bradyzoite transcription were notable. (1) The small heat shock protein, *BAG1* was the first bradyzoite-specific marker characterized[30–32] and has served as a universal marker for over 30 years. It was therefore unexpected that over half of Group B bradyzoites lacked detectable expression of *BAG1* mRNA (Fig. 6a). By contrast, Group B bradyzoites as well as bradyzoite Groups A, C & D and to a lesser extent Group E bradyzoites expressed high levels of *LDH2* mRNA (Fig. 6a; Fig. S4A) supporting *LDH2*, a lactate dehydrogenase, as a more reliable marker for in vivo bradyzoites. (2) The SRS22 family of surface proteins are primarily merozoite-specific[23,24] and Group B bradyzoites expressed two SRS22 members, *SRS22A* as mentioned and also *SRS22i* mRNA (Fig. 6b). The analysis of the Group B bradyzoites lacking BAG1-identified two other SRS22 family members, *SRS22E* and *SRS22G*, although the expression of these SRS22 family members were lower than *SRS22A* and *SRS22i* (Supplementary Data 1). (3) Group B bradyzoites also had the highest levels of 24 dense granule protein (GRA) mRNAs including bradyzoite-specific *GRA46* and *GRA63* (Fig. 6a) as well as merozoite GRA proteins *GRA11A* and *GRA11B* and *GRA80-82* (Fig. S4A, B). (4) Finally, Group B bradyzoites expressed significantly higher levels (e.g. 21-fold higher Group B vs D, Supplementary Data 1) of merozoite transcription factor, *AP2VIIa-1* (Fig. 6c)[24]. Altogether, these features may indicate Group B bradyzoites are poised to initiate merozoite development in the cat. Further study will be required to understand whether Group B bradyzoites or another in vivo bradyzoite group are the primary drivers of bradyzoite-to-merozoite development.

In addition to the distinct signature of Group B *SRS22A* expression, the in vivo bradyzoite Groups A-E displayed other distinctive SRS mRNA profiles (Fig. 6b). Ten SRS protein mRNAs had peak levels in Group E bradyzoites (Fig. 6b, Supplementary Data 1), including four merozoite-specific SRS proteins, SRS16E, 23, 30 C, 53 F and one SRS mRNA encoding a tachyzoite-associated protein SRS20A. The SRS mRNA profiles of bradyzoite Groups C and D were partially shared reflecting the nearly 70% overlap of Group C transcripts with the transcriptome of Group D bradyzoites. In particular, the expression of the *SRS49* gene family mRNAs (*SRS49A-D*) was a distinct feature of bradyzoite Groups C and D (Fig. 6a, b). Group D bradyzoites had the highest proportion of unique mRNA expression (Fig. 5b, 1074 mRNAs) and also one of the lowest levels of *SRS22A* expression indicating Group D is likely a major source of SRS22A- bradyzoites. Intriguingly, the highest levels of *SRS9* mRNA expression were in Group C and D, which correlates with the SRS22A- bradyzoite phenotype (see Fig. 3). The results above demonstrated SRS22A- bradyzoites initiate bradyzoite-to-bradyzoite development and it was SRS22A- bradyzoite infections that were responsible for spontaneous cyst wall formation (Fig. 3 and[13]). Consistent with these findings, Group C and D bradyzoites expressed the highest levels of SRS mRNAs that encode major cyst wall proteins, with 75% of bradyzoites expressing *SRS44 (CST1)* (Fig. 6a) and 62% of bradyzoites expressing SRS13 (Fig. S4A).

Group A bradyzoites express the 3rd highest number of unique transcripts (Fig. 5b, 172 mRNAs) including many hypothetical proteins and increased expression of selective rhoptry and RON mRNAs (Fig. 6a, Fig. S4C). This characteristic partially overlaps with Group E, which also shows increased *BRP1* and bradyzoite-specific ROP transcripts (e.g. *ROP28, 30, 42, 43; RON2L1*, Fig. S4A)[23,24,33] and the Group A-enhanced *ROP kinase (TGME49_308093*, Fig. S4)). In Group A, the strong upregulation of rhoptry mRNAs (e.g. >50-fold for some), is not due to S/M cell cycle enrichment[34], unlike Group E, where rhoptry expression correlates with S/M transcripts (Fig. S4C, A vs E heat map). Both Group A and E display elevated *AP2XII-9* (Fig. 6a), a known activator of rhoptry and RON genes[35,36]. We do not fully understand the biological role for increased rhoptry gene transcription in Group A bradyzoites; however, our previous cell cycle studies of rhoptry proteins in tachyzoites[37] demonstrated the transcription of this class of mRNAs was uniquely sensitive to intercellular egress and invasion, which might indicate Group A bradyzoites were preparing to egress prior to tissue cyst harvest.

Finally, trajectory analysis, although not definitive, supports the presence of alternative bradyzoite developmental programs as seen with SRS22A+ and SRS22A- parasites. The inferred trajectory begins with parasites in Group D, progresses toward the transitional Group C/D population, and then bifurcates towards either Group B (SRS22A + ) or A (Fig. S5B).

These scRNA-seq data support the SRS22A and SRS9 findings that cysts contain distinct subsets of bradyzoites that lead to predictable patterns of development. These five populations separate based, not on cell cycle, but signatures that include characteristic proteins involved in parasite invasion and developmental regulation suggesting

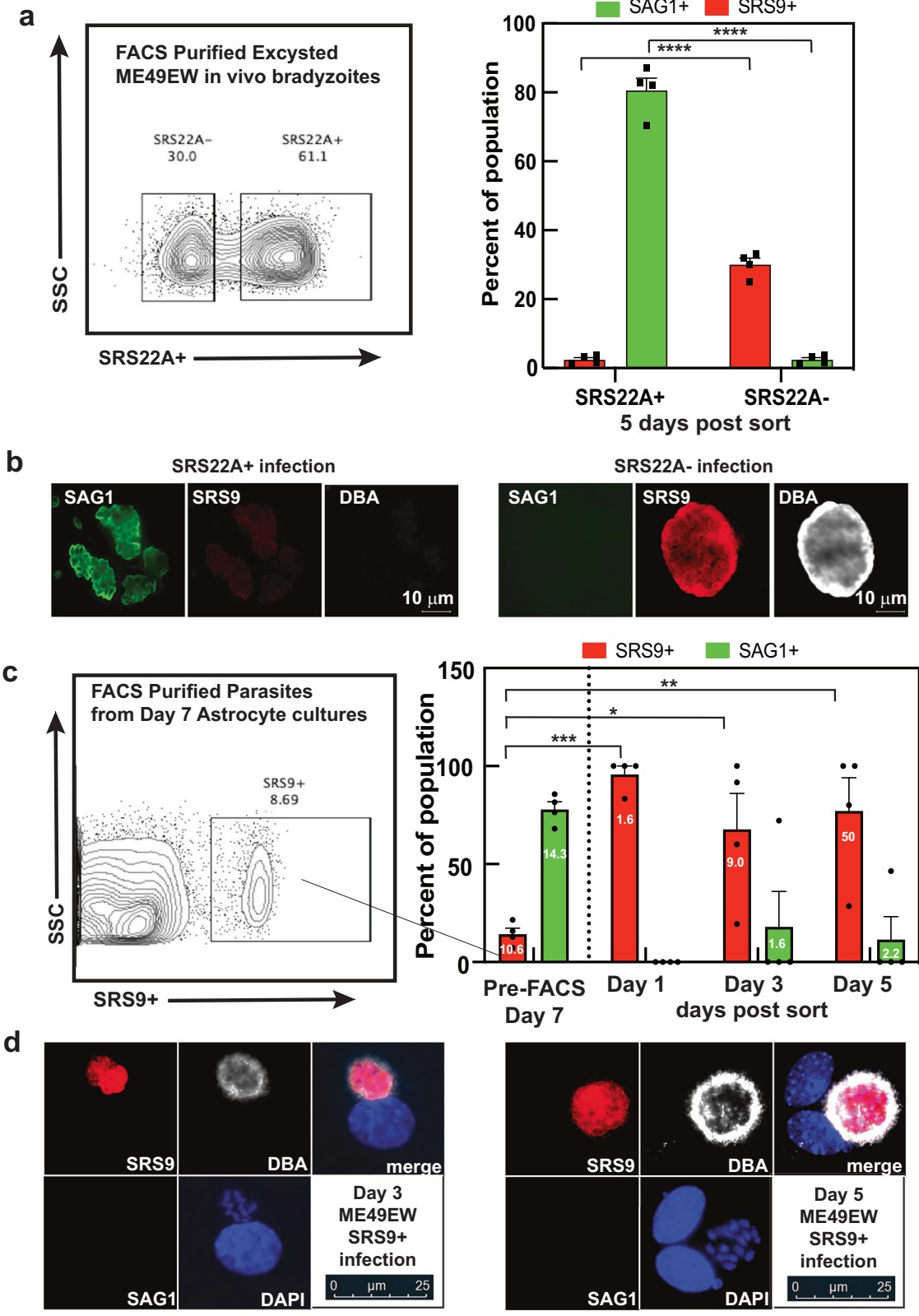

some of these populations are precursors to merozoites, tachyzoites and renewal of the bradyzoite population.

## Discussion

In these studies, we describe three major advances in our understanding of bradyzoite biology. Firstly, the expression of the SAG related protein, SRS22A, occurs in bradyzoites formed in vivo, is lost within 24 h of in vitro culture and defines a population that transforms to the fast replicating, disseminating tachyzoite. Secondly, the alternative population found following recrudescence, brady-brady replicating parasites, represents a SRS22A- population that is stable and maintains the propensity to replicate as a bradyzoite (Fig. 7 model). Lastly, using scRNAseq of excysted bradyzoites, we observed further bradyzoite heterogeneity represented by five main groups of parasites

**Fig. 3 | Developmental fate of SRS22A bradyzoite subtypes. a** Excysted ME49EW bradyzoites were isolated from CBA mice (40-day post-infection) and purified by FACS into SRS22A+ and SRS22A- populations (Fig. S2A). Pre-sort populations consisted of 63% SRS22A+ and 36% SRS22A- bradyzoites. Primary mouse astrocytes were inoculated with SRS22A+ or SRS22A- bradyzoites (500 parasites/cover slip), and co-stained for SRS9 and SAG1 at 5-days post-infection. Total number of parasites was determined for each coverslip (3.1-fold increase for SRS22A + , 1.1-fold increase for SRS22A-). Three replicate counts of end-stage SAG1+ and SRS9+ only vacuoles were performed. Data are mean ± SEM. Statistical analysis: two-way ANOVA with Sidak's multiple comparisons (main effects; SRS22A status: $P < 0.0001$, $F = 15.5$, and SRS22A+ *vs.* SRS22A- populations: *P*-value < 0.0001 for both SAG1 and SRS9). **b** Cyst wall formation (DBA + ) was observed only in SRS9+ bradyzoites that were the major product of the SRS22A- infections. Scale bar = 10μm. FACS sorting were repeated 4-5 times (Fig. S2B). **c** SRS9 + /SAG1- parasites from ME49EW Day-7- recrudescing populations[13] were purified by FACS and used to infect primary astrocytes. Day-7 ME49EW parasites do not express SRS22A (Fig. 2a). Triplicate counts of vacuole size (column numbers indicate mean size) and SAG1+ or SRS9+ only antigen expression were quantified at 1-, 3-, and 5-days post-infection (no SAG1+ only vacuoles at Day1). Vacuoles co-expressing both antigens comprised 4-14% of total vacuoles at each timepoint (not graphed). Mean vacuole size and end-stage fractions of the pre-sort Day-7-population are indicated. FACS sorting were repeated in four independent experiments (Fig. S2D). Data are mean ± SEM. Statistical analysis: two-way ANOVA with Sidak's multiple comparisons (Figs. 3a, c). Main effect (SRS9 status): $P = 0.0002$, $F = 20.30$, multiple comparisons: start *vs.* day 1 ($P = 0.0002$), day 3 ($P = 0.0116$), day 5 ($P = 0.0029$). **d** Representative images of day-3 and −5 post-infection vacuoles in astrocytes infected with purified SRS9+ bradyzoites. Scale bar = 25 μm. ****$P < 0.0001$, ***$P < 0.001$, **$P < 0.01$, *$P < 0.05$. Source data are provided as a Source Data file for this figure.

identifiable by their expression of AP2 transcription factors and SRS proteins.

Our earlier work showed that sporozoite and bradyzoite infections of host cells follow a shared developmental sequence[13,14,38,39]. Parasites first switch to a fast-growing tachyzoite stage, then shift almost synchronously to slower tachyzoite growth 6-7 days later, after which bradyzoites arise. A return to the fast-growing tachyzoite has never been observed, and this fast-tachyzoite form appears only in ex vivo sporozoite and bradyzoite infections. The reason for this restricted developmental context[13] and molecular basis of the origin of the fast-growing tachyzoites remains unknown. In this study, we advanced our understanding of the ingredients needed to produce natural fast-growing tachyzoites. Infection of astrocytes with FACS-purified ME49EW SRS22A+ bradyzoites from infected mice clearly established that fast-growing tachyzoites originate from the SRS22A+ bradyzoite population (Fig. 3). We show here that switching parasite cell culture to an alkaline media was incapable of inducing the formation of SRS22A+ bradyzoites (Fig. 2), and this explains why fast-tachyzoites have not been observed in the most commonly used in vitro model of bradyzoite development. Additionally, the experiments here demonstrate SRS22A expression is not a good marker of in vitro bradyzoite development as this surface protein is rapidly lost within 24 hours of infecting primary astrocytes with bradyzoites from mice (Fig. 2b). Recent studies have shown that terminally differentiated human myotube (KD3) cultures support tissue cyst formation without the need for alkaline-stress following infection with tachyzoites and can also maintain bradyzoites in long-term culture[40,41]. Future work will be needed to fully investigate SRS22A+ expression in the variety of in vitro models including the myotube model. However, it should be noted that SRS22A mRNA levels were extremely low (<37th percentile, ToxoDB) in an infection-study of four different cell types including murine neurons, fibroblasts, skeletal muscle cells, and astrocytes[42]. Our recent work showed that ME49EW fast-growing tachyzoites disseminate more efficiently in mice[13], suggesting a key role for this stage in the intermediate host life cycle. Parasite development in the intermediate hosts is often viewed as a linear construction[43] in which tachyzoites convert stepwise into a single end-stage, mature bradyzoite, with bradyzoite- or sporozoite-initiated infections differing only by an initial switch back to the tachyzoite stage. However, our recent studies of bradyzoite recrudescence[13] discovered spontaneous bradyzoite replication alongside the more prevalent tachyzoite vacuoles in astrocytes[13], challenging this simple linear model. Consistent with our findings, previous studies have shown that tissue cysts are heterogenous and that bradyzoite replication is dynamic over time[11]. Similarly, cyst heterogeneity extends to differences in mitochondrial morphology, and amylopectin accumulation, which undergo cyclical changes during chronic infection[44,45]. Another study found that BAG1+ bradyzoite-infected HFFs form two distinct transcriptional subsets, suggesting that variability in host responses may reflect heterogeneity among the bradyzoites[46]. We now

understand that the origin of these distinct replicating stages was not a single parent, but instead two distinct parents present in our ex vivo inoculum. The SRS22A- bradyzoite subtype was the parent of bradyzoite-to-bradyzoite tissue cysts and the SRS22A+ bradyzoite subtype was responsible for the dominant bradyzoite-to-tachyzoite recrudescence. Thus, bradyzoite development is not linear but rather the result of multiple active pathways progressing in parallel. This raises the question of how many bradyzoite subtypes there are given that bradyzoites initiate at least three different pathways: brady-brady, brady-tachy (and the reverse) and brady-merozoite (Fig. 7 model). Considering the growth patterns of the SRS22A+ and SRS22A- bradyzoites (Fig. 1), subtypes are formed early, and tissue cysts are heterogenic with the expression of SRS22A. This heterogeneity is established within two weeks of infection and maintained throughout the chronic infection. This points to a desired characteristic of biology to have diversity at all levels and specifically in *Toxoplasma* infection of having diversity of bradyzoites within and between cysts. Moreover, the SRS22A subtypes (+ or -) are stable, can be purified and the developmental preferences are active in cell culture and in mice. This developmental arrangement would afford individual tissue cysts the ability to successfully transmit *Toxoplasma* infection to a wide range of animal hosts. Our scRNA-sequencing of in vivo bradyzoites suggest up to 4-5 bradyzoite subtypes within a single tissue cyst. We do not fully understand how bradyzoite subtypes are regulated, although it is likely that epigenetics plays a major role, and it is clear that in vivo bradyzoite groups A-E express distinct ApiAP2 factors (Fig. 6c). Supporting these concepts, recent studies of ME49EW mutant bradyzoites demonstrated that genetic knockout of ApiAP2 transcription factor, *AP2IX-9 (TGME49_306620)*, or the bradyzoite cyclin, *TgCYC5 (TGME49_293280)*, disrupted the balance of SRS22A+ versus SRS22A- cyst subtypes, which in turn caused shifts in bradyzoite recrudescence commensurate with the altered SRS22A phenotype in each bradyzoite mutant[47].

The heterogeneity of SRS22A+ bradyzoites (Fig. 3) was confirmed at the mRNA level with Group B bradyzoites the major source of SRS22A+ expression (Fig. 6). Group B bradyzoites have a unique and unexpected phenotype with >50% of these bradyzoites lacking expression of BAG1. This is consistent with BAG1 being dispensable for bradyzoite development[31,48–50]. Group B bradyzoites also had low expression of *SRS9* and *CST1 (SRS44)* among other groups and are the drivers of differentiation into fast-replicating tachyzoites (Figs. 3 and 6). Group B bradyzoites had the highest expression of *SRS22A* mRNA, dense granule proteins *GRA11*, and *GRA80-82*, which are all merozoite-specific proteins. Turning off BAG1 expression may signal a priming event that pre-stages SRS22A+ Group B bradyzoites to recrudesce into the tachyzoite or merozoite stage depending on the host environment. Consistent with this, epigenetic priming events were proposed to be responsible for stage transition. For example, one hypothesis involves regulation of chromatin remodeling by the MORC/HDAC3 repressor complex[29,51]. Our first encounter with priming of

**a** **Tachyzoites (SAG1+) in PECS at 5-days post-infection** Infected PECS at 5-days post-infection

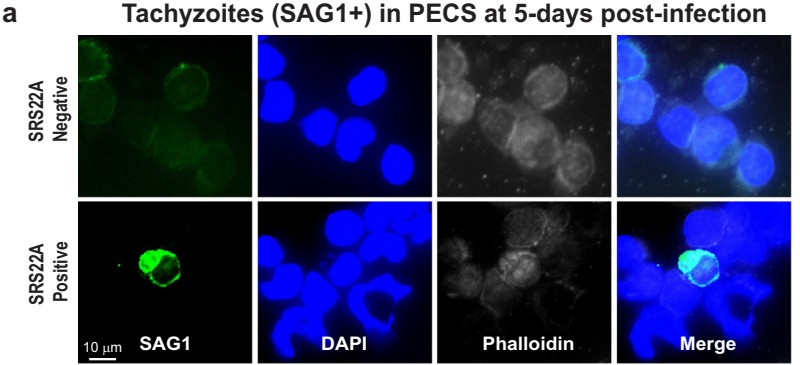
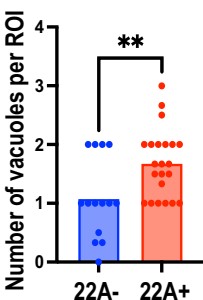

**b Parasite burden from SRS22A mouse infections (5-days post-infection)**

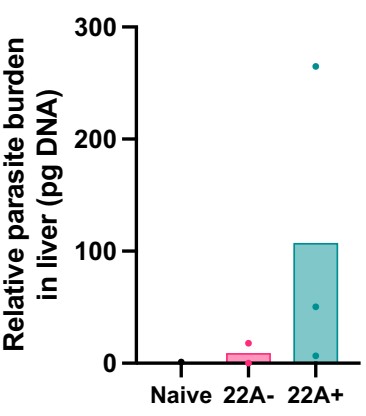
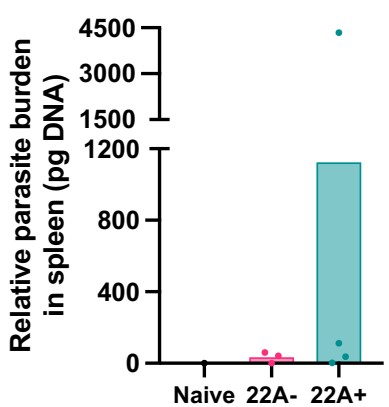
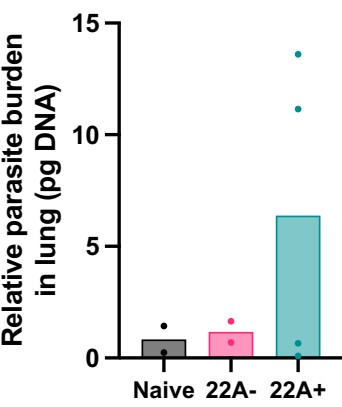

**c**

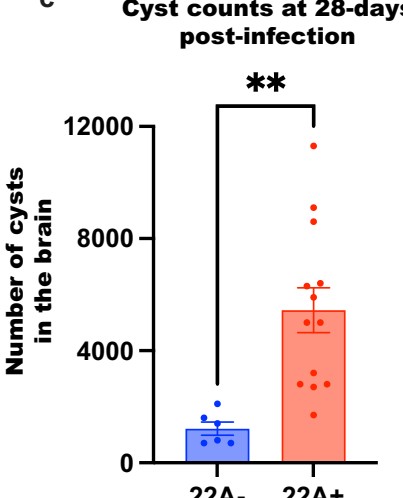

developmental gene expression was studying *Eimeria bovis* sporozoites and merozoites indicating this phenomenon is an ancient property of Coccidian parasites[52]. Both tachyzoites and merozoites lack BAG1 expression and they are the major replication stages for their respective life cycles. Consequently, they have shared metabolic needs and derive energy through similar biochemical mechanisms[23]. Supporting the premise of a common parent are recent studies that demonstrate altering a single AP2 factor[53,54] can switch parasite

replication from endodyogeny of the tachyzoite to endopolygeny of the merozoite[55]. Further studies to investigate the functional breadth of Group B bradyzoites are underway.

We showed above that SRS22A- bradyzoites are the primary initiator of bradyzoite replication and it is in these vacuoles that spontaneous cyst wall formation occurs in ex vivo infected astrocytes ([13], Fig. 3). scRNA-sequencing identified Groups C and D as the major source of SRS22A− bradyzoites (Fig. 6) and exhibit the highest *SRS9*

**Fig. 4 | In vivo studies of FACS purified SRS22A+ and SRS22A- bradyzoites.**
**a** Representative images of peritoneal exudate cells (PECs) at 5-days post-infection (DPI). PECs were collected from each mouse by injecting intraperitoneally with 5 ml cold PBS, and 100,000 cells per sample were deposited onto microscope slides using cytospin funnels. Right panel: quantification of the number of SAG1+ (green color) parasite vacuoles per region of interest (ROI). Data are as mean ± SEM; $n = 3$ mice per group. Statistical significance was assessed using unpaired $t$-test (two-tailed).**$P < 0.01$. **b** Parasite burden in the liver, spleen and lung was assessed by qPCR targeting the *Toxoplasma B1* gene. Liver parasite burden: $n = 1$ (naïve), $n = 2$ (22 A⁻), $n = 3$ (22 A⁺), spleen parasite burden: $n = 1$ (naïve), $n = 3$ (22 A⁻), $n = 4$ (22 A⁺), and lung parasite burden: $n = 2$ (naïve), $n = 2$ (22 A⁻), $n = 4$ (22 A⁺) mice per group.

**c** Mice were infected intraperitoneally with 10,000 FACS-purified SRS22A+ or SRS22A- bradyzoites. At 28-days DPI, brain cysts were enumerated from whole brain homogenates. Homogenization was performed sequentially using 18 G, 20 G, and 22 G needles. Data are presented as mean ± SEM; $n = 6$ mice for 22 A⁻ and $n = 13$ for 22 A⁺ infections. Representative immunofluorescence images of cysts collected at 28-days post-infection from the infections described in (A). Brain homogenates were smeared onto microscope slides, fixed in methanol, and stained with anti-SRS22A antiserum. and $n = 24$ cysts (SRS22A⁺ infections). Statistical significance was assessed using unpaired t-tests. **$P < 0.01$. Source data are provided as a Source Data file for this figure.

mRNA *CST1* and *SRS13* mRNA levels, consistent with their trajectory toward cyst formation. Like BAG1, SRS9 does not appear to be a canonical marker of in vivo bradyzoite development and the preference of SRS9 expression in SRS22A- bradyzoites from mice explains why purified SRS9 + /SRS22A- bradyzoites from astrocytes phenocopied the developmental course of in vivo SRS22A- bradyzoite recrudescence (Fig. 3b). Previous work[13] suggested that slower-replicating bradyzoites that emerge post-recrudescence (brady-brady) maintain long-term cysts rather than driving tissue dissemination during the acute stage of infection. The infection of mice with purified SRS22A+ and SRS22A- bradyzoites supports this concept: SRS22A+ infections produced more SAG1+ parasites in PECS during acute infection, which was later reflected in more cysts in the chronic phase as opposed to infections with purified SRS22A- bradyzoites (Fig. 7).

The role of Group A bradyzoites remains unclear. Our RNA-sequencing reveal they upregulate many rhoptry effector mRNAs (Fig. S4C), a gene class previously found to be highly responsive to egress and invasion cues[37]. *Toxoplasma* can invade many host cells without establishing a parasitophorous vacuole or tissue cyst[56,57], suggesting Group A bradyzoites may be preparing to egress or primed for host cell invasion. Bradyzoite motility is just as effective as tachyzoites for invasion[58–60]. Intriguingly, we have found that the deletion of bradyzoite-specific cyclin, TgCYC5, may cause an increase in these bradyzoite dissemination events[47]. Further work is needed to determine whether this cyclin mutant have an enriched Group A transcriptome.

Several genes have been linked to bradyzoite differentiation pathways, including *BFD1, BSM and BCLA*[28,61]. However, low *BFD1* expression, and uniformly high BSM expression across all clusters limit their use to distinguish *BFD1*-dependent or independent populations. *BCLA*, associated with cysts, shows the strongest differential expression and is largely absent from Group B bradyzoites (Fig. S5A; GEO Accession: GSE311669). The scRNA-seq datasets generated here will be valuable for further investigations of *Toxoplasma* strains that vary in their cyst forming capacity and infection outcomes.

Overall, our findings revealed there are multiple subtypes of bradyzoites with distinct transcriptomes supporting the idea that tissue cysts are not homogenous and that a linear model of cyst maturation is unlikely. Furthermore, we identified novel bradyzoite markers that add to the current canonical markers and help distinguish natural bradyzoites from those generated in vitro. The data provides evidence as to why repeated culture and passage has led to vast variation in *Toxoplasma* strains used by the community. This puts the tissue cyst at the center of the *Toxoplasma* life cycle generating the flexibility required to maintain a chronic infection, initiate new infections and likely the adaptability to successfully invade multiple mammalian hosts. These findings open new avenues for investigating native bradyzoite biology and tissue cyst diversity.

## Methods
### Parasite maintenance and mouse infections
Our research complies with all relevant ethical regulations (Biological Use Authorization, BUA-R202 approved by the Institutional Biosafety

Committee at Environmental Health and Safety Office at the University of California, Riverside). All animal experiments (Protocol # 108) were approved by the Institutional Animal Care and Use Committee (IACUC) of the University of California, Riverside. Six-week-old female CBA/J (strain #: 000656), C57BL/6 J (strain #: 000664) and Swiss Webster (SWR/J) (strain #: SW-F, Taconic Biosciences) mice were obtained from Jackson Laboratories (Jackson ImmunoResearch Laboratories, Inc., West Grove, PA, USA) and Taconic Biosciences (Rensselaer, NY, USA) and maintained in a pathogen-free vivarium on a 12 h light/12 h dark cycle at 20–22 °C room temperature with ad libitum access to water and food. Type II ME49EW parasites were propagated in vivo by alternative passage between resistant (Swiss Webster, SWR/J) and sensitive (CBA/J) mouse strains[47]. Mice were infected via intraperitoneal injection with 10 tissue cysts (in 200 μl) prepared from brain homogenates of mice infected for 30 days. It has previously shown that it takes about a week for tissue cysts to appear in mouse infections[62]. Consistent with this, we have previously shown that excysted bradyzoites differentiate into fast-growing tachyzoites and then switch to a slower growth phase and reform bradyzoites after day 6[13]. Considering these findings, we did not examine tissue cysts earlier than 14 days post-infection. For some experiments we needed a stable tachyzoite source for comparison purposes or to duplicate the conventional alkaline-stress model (Figs. 1a, 2a). In these experiments, we used the genome reference strain Type II ME49B7, which has been adapted to in vitro cell culture. A continuous replicating tachyzoite form for in vitro experiments is not available for the ME49EW strain as it is maintained exclusively in vivo. We also used tachyzoites of the Type I RH strain to check the bradyzoite-specificity of the anti-SRS22A antiserum in Western analysis (Fig. 1b).

### Bradyzoite purification and ex vivo cultures following recrudescence
Following 40-day post-infection, ME49EW bradyzoites were isolated from infected mouse brains (CBA mice) via Percoll gradient[13]. Neonatal (from C57BL/6 J mouse pups) cortical astrocytes grown in T-75 cm2 vented flasks were used for ex vivo parasite cultures initiated by excysted bradyzoites. Primary astrocyte cultures were inoculated with excysted bradyzoites or day 3 and 5 parasite cultures at an MOI of 0.5 as previously described[13]. Infected cells were cultured in DMEM media supplemented with 5% heat-inactivated fetal bovine serum (FBS), 1% penicillin-streptomycin (Genclone), 1% GlutaMax supplement (Thermo Fisher Scientific, Waltham, MA, USA), 1% Na pyruvate and 2.5% HEPES buffer (Thermo Fisher Scientific Waltham, MA, USA) and placed in humidified hypoxia chambers containing 5 % oxygen and 5 % $CO_2$ (nitrogen balanced).

### Production of antiserum against SRS22A
Rabbit antiserum was generated in-house against the *Toxoplasma* protein SRS22A using a selected peptide sequence (Fig. S1A, *LRGNDGRSSRVIEKEAEVAK*). Rabbits were immunized with the peptide, and antisera were collected at 28 days (first bleed), 56 days (second bleed), and 72 days (third bleed) post-immunization (Thermo Fisher Scientific Waltham, MA, USA). The peptide (20 amino acids) selected

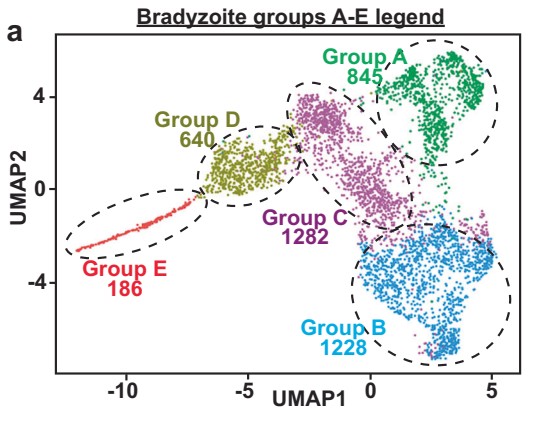

**a** Bradyzoite groups A-E legend

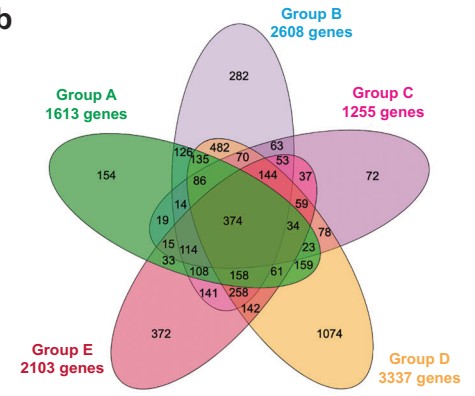

**b**

Group A 1613 genes
Group B 2608 genes
Group C 1255 genes
Group E 2103 genes
Group D 3337 genes

**c**

Bradyzoite groups A-E transcribed genes (<0.05 p-values) compared to the published G1 and S/M subtranscriptomes

| Subtype | G1 SubT (1566 genes) | S/M SubT (1148 genes) | Non-CC genes | Total genes |
|---|---|---|---|---|
| Group A | 418(24%) | 511(30%) | 773 | 1702 |
| Group B | 620(23%) | 701(26%) | 1375 | 2696 |
| Group C | 251(18%) | 448(33%) | 676 | 1350 |
| Group D | 856(24%) | 573(16%) | 2091 | 3520 |
| Group E | 333(15%) | 745(34%) | 1097 | 2175 |

**d Five volcano plots**

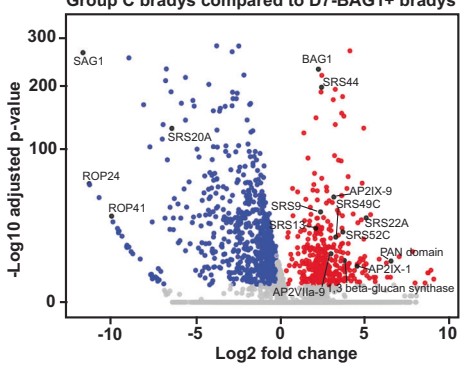

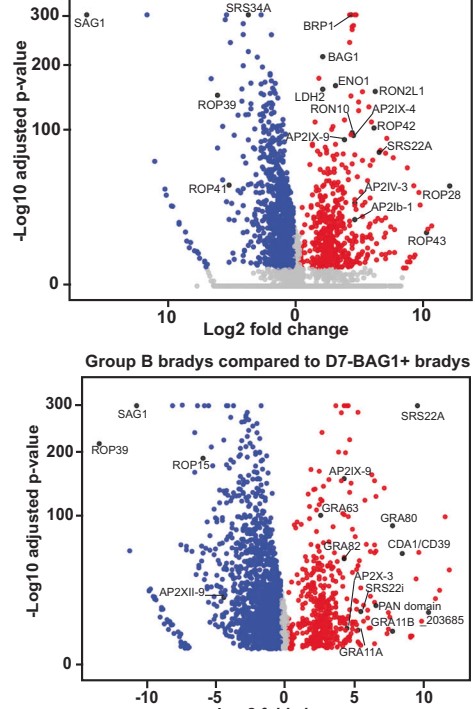

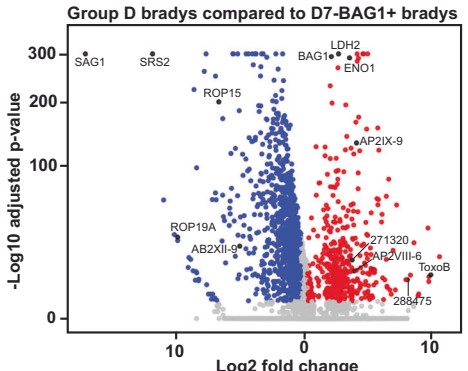

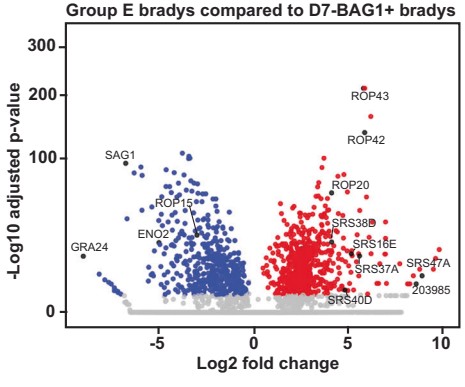

for production of anti-SRS22A antisera was confirmed by Protein BLAST to be uniquely present in the SRS22A protein sequence among SRS family and other proteins encoded in the *Toxoplasma* genome.

**FACS sorting of bradyzoite populations**

Upon excystation of purified ME49EW cysts, at least one million bradyzoites were stained for flow cytometry analysis using specific antibodies against SRS9 or SRS22A. Briefly, parasites were incubated with rabbit anti-SRS9 (kindly provided by John Boothroyd, Stanford University) or rabbit anti-SRS22A (generated in-house). Primary antibody incubation was performed at a 1:50 dilution in PBS in 5 ml round-bottom FACS tubes for 30 min at room temperature. Following incubation, each sample was washed with 1 ml of sterile PBS and centrifuged at 450 g for 5 min. Secondary antibodies were diluted 1:1000

**Fig. 5 | Single cell RNA-sequencing of in vivo bradyzoites. a** UMAP projection of individual in vivo bradyzoites from 30-day infected CBA mice shows five potential subtypes; Groups A, B, C, D, and E with the number of bradyzoites analyzed indicated below each group label. **b** A Venn Diagram of significant gene expression ($P < 0.05$) from bradyzoite Groups A-E was generated. This analysis shows all common, partially shared and unique genes associated with each Group. **c** Table indicates the total number of genes in Groups A-E that are significantly expressed (Supplementary Data 1, $P < 0.05$) and the breakdown of shared expression with the published G1 and S/M sub-transcriptomes[14,27,37,63] as well as the number of non-cell cycle genes. **d** Volcano plots for each Group A-E were generated in comparison to BAG1+ bradyzoites in day 7 ex vivo cultures of infected astrocytes. Grey indicates genes with $p > 0.05$, red indicated significantly upregulated genes ($P < 0.05$) and blue indicates downregulated genes ($P < 0.05$). Up to 15 upregulated and 5 downregulated genes are indicated on each volcano plot. Differential expression was assessed using Seurat's Wilcoxon rank-sum test (FindMarkers), with Benjamini-Hochberg correction for multiple testing. Results are presented as volcano plots of log2 fold change versus -log10 adjusted $p$-value. Source data are provided as a Source Data file for this figure.

in PBS, with Alexa Fluor 647 goat anti-rabbit IgG (A-21244, Lot: 2527970, Invitrogen, Carlsbad, CA, USA) used for SRS9 and Alexa Fluor 488 goat anti-rabbit IgG (A-11008, Lot: 1678787, Invitrogen, Carlsbad, CA, USA) used for SRS22A. Parasites were then resuspended in 50 μl of diluted secondary antibodies and incubated for 30 min at room temperature in the dark. After incubation, stained parasites were washed with 1 ml of PBS, centrifuged, and resuspended in 300 μL of PBS for FACS analysis. Flow cytometry and cell sorting were performed using the MoFlo Astrios EQ Cell Sorter (Beckman Coulter, Brea, CA, USA) at the UC Riverside School of Medicine Flow Cytometry Core Facility to identify and sort distinct SRS9+ or SRS22A+ parasite populations. SRS9 and SRS22A expression were detected in the APC and FITC channels, respectively, and parasite populations were collected in 1 ml of infection media. Primary astrocyte cultures grown on coverslips were infected either with purified SRS22A+ or SRS22A− sorted parasites (500 parasites per well in 6-well plates, approximately MOI of 0.0004) or with SRS9+ parasites (2500 parasites per well in 24-well plates, approximately MOI of 0.01). Full gating strategy, controls and reproducibility data are shown in Fig. S2A-D. FACS purification of the SRS22A+ and SRS9+ populations was highly reproducible as evidenced by low variability obtained from separate experiments run on different dates (Fig. S2B-D).

### Immunofluorescence staining

Population growth and SRS antigen expression patterns (SAG1 vs. SRS9) were assessed in primary astrocytes infected with sorted parasite populations on Day 5 post-infection. Cells were fixed in 4% paraformaldehyde (PFA), followed by three 1 min washes with PBS. After permeabilization in acetone for 10 min, cells were washed three more times with PBS for 1 min each. Next, cells were blocked with 5% donkey serum in PBS and incubated with antibodies specific for bradyzoites and tachyzoites, including rabbit anti-SRS9 (1:1000 dilution), mouse anti-SAG1 (1:1000 dilution, catalog # 9070-2020, Lot: 162026, Bio-Rad, Hercules, CA,USA), and biotinylated DBA (1:500 dilution, B-1035-5, Lot: ZJ1117, Vector Laboratories, Newark, CA, USA) to identify the cyst wall. Primary antibody incubation was carried out overnight at 4 °C. The following day, cells were washed three times with PBS for 5 min each and incubated with secondary antibodies for 1 h at room temperature. The secondary antibodies used included donkey anti-rabbit Alexa Fluor 568 (1:1000 dilution, A10042, Lot: 1964370, Invitrogen, Carlsbad, CA, USA), goat anti-mouse Alexa Fluor 488 (1:1000 dilution, A-11029, Lot: 2821059, Invitrogen, Carlsbad, CA, USA), and Alexa Fluor 647-streptavidin conjugate (1:1000 dilution, S32357, Lot: 2179341, Sigma-Aldrich, St. Louis, MO, USA) to detect rabbit anti-SRS9, mouse anti-SAG1, and biotinylated DBA, respectively. After incubation, cells were washed three times with PBS for 10 minutes each and rinsed once with autoclaved water. DAPI VIBRANCE (Vectashield Vibrance, H-1800, Vector Laboratories, Newark, CA, USA) was then applied to each coverslip before mounting onto a glass slide. Parasite vacuoles were counted using a Leica DMI 6000 immunofluorescence microscope with a 40X objective, and vacuoles were enumerated.

### Single-cell RNA-sequencing of in vivo bradyzoites

Single-cell RNA sequencing (scRNA-seq) was performed on excysted bradyzoites and parasite cultures obtained from ex vivo Day 7 astrocytes at a sequencing depth of one billion reads[13]. This was achieved using the 10X Genomics Chromium Next GEM Single Cell 3' Reagent Kit v3.1 (10X Genomics, Inc., Pleasanton, CA, USA)[13]. Sequencing was conducted at the San Diego IGM Genomics Center using an Illumina NovaSeq 6000. As before, Cell Ranger version 5.0.1 was used for read alignment to a custom reference genome (ME49 *Toxoplasma gondii* reference genome) and for count matrix generation. The scRNA-seq data were embedded using Uniform Manifold Approximation and Project (UMAP) and parasite transcriptome intersection determined using the Seurat package (R version 4.4.2 and Seurat 5.1.0). scRNA-seq data were analyzed using the Seurat package (R version 4.4.2 and Seurat 5.1.0) with libraries 'ggplot2', 'tidyverse', 'conflicted', 'gridExtra', 'dplyr', 'EnhancedVolcano' and 'ggrepel'. Dimensionality reduction was first performed using Principal Component Analysis (PCA), followed by construction of a Shared Nearest Neighbor (SNN) graph via the 'FindNeighbors' function. Cells were then clustered using the Louvain algorithm at a resolution of 0.2, yielding five transcriptionally distinct clusters. For visualization, cells were embedded in two dimensions using (UMAP).' A non-parametric Wilcoxon rank test was applied to identify differentially expressed genes among Groups A-E. Volcano plots were generated from a comparison of each bradyzoite Group A-E (chronic phase infection) and the BAG1+ bradyzoites (early phase) that spontaneously arise in our ex vivo bradyzoite recrudescence model at Day-7 as explained in Results.

Trajectory analysis was performed using the dyno R package (dyneverse.org). We followed the recommendations of the Guidelines Shiny application to select trajectory-inference methods appropriate for our dataset. Based on these guidelines, we implemented four recommended methods including Slingshot, PAGA, PAGA-tree, and MST.

### Bradyzoite differentiation assay under alkaline stress

Alkaline stress assay was performed in HFFs (ATCC SCRC-1041)[63,64]. A bradyzoite differentiation media was prepared using RPMI 1640 (GIBCO) medium supplemented with 5% FBS, 50 mM HEPES, 100 unit/ml penicillin, and 100 μg/ml streptomycin. The alkaline medium pH was adjusted to 8.2 using NaOH. Briefly, HFFs were infected with ME49B7 parasites for 4 h, and then media was removed and rinsed four times with parasite media (5% FBS, 1% Gln, 100 unit/ml penicillin, and 100 μg/ml streptomycin) to remove unattached parasites. The bradyzoite differentiation media was added onto the cells and cell culture flasks were placed at 37 °C for 48 h. Cells were fixed in 4% PFA post-infection and then stained for SAG1, SRS9 and SRS22A using immunofluorescence staining as described above.

### Preparation of PECs and brain homogenates at days 5- and 28-post-infection

On 5-days post-infection, CBA mice were anesthetized and injected intraperitoneally with 5 ml of cold PBS with 1% BSA. Peritoneal exudate cells were collected by withdrawing the injected solution and subsequently enumerated. For each mouse, 100,000 cells were deposited onto microscope slides using cytospin funnels and centrifugation at 300 *g* for 5 min. On 28-days post-infection, infected mouse brains were harvested in 3 ml cold PBS, and brain homogenates were prepared by sequentially passing the tissue through 18 G, 20 G, and 22 G needles. A

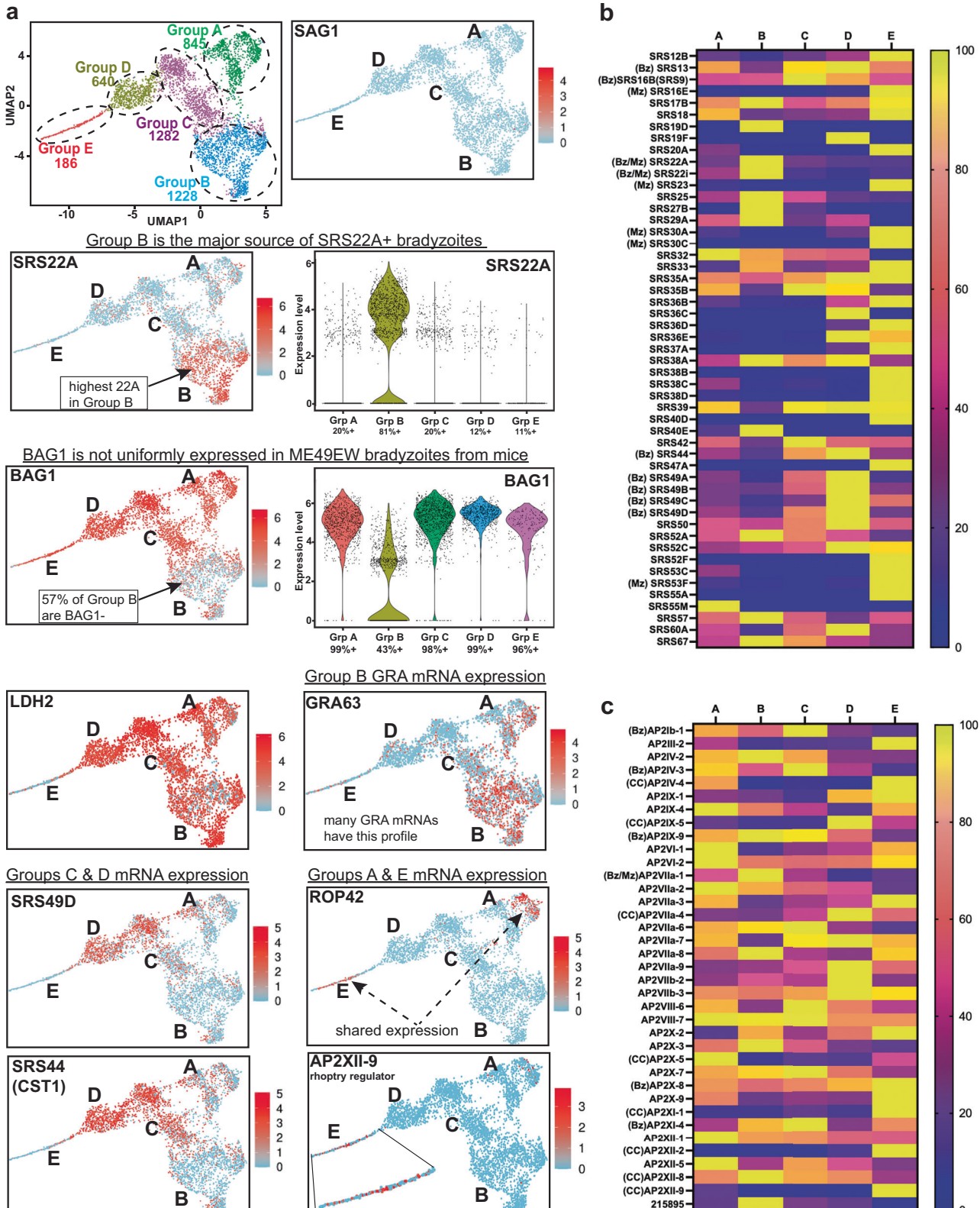

**Fig. 6 | Gene expression profiles of five bradyzoite subtypes. a** (top row) The left UMAP image of individual in vivo bradyzoites that clustered in Groups A-E is included as a reference. UMAP image of SAG1 expression shading is on the right. (second row) UMAP projection and violin plot of SRS22A expression highlights expression in Group B bradyzoites (81% of all bradyzoites expressing SRS22A). (third row) UMAP projection and violin plot of BAG1 shows non-uniform expression among the clusters with Group B having the least expression of BAG1 mRNA. (fourth row) UMAP image of LDH2 and GRA 63 expression. Violin plot of LDH2 expression and a heat map of dense granule gene expression is shown in Fig. S4. (left UMAP images in rows five and six). Groups C and D shared expression of SRS49D and SRS44 (CST1). (right UMAP images rows five and six) Shared expression of ROP42 and AP2XII-9 in Groups A and E. Group E expression image of AP2XII-9 is magnified. **b, c** SRS and ApiAP2 heat maps of normalized mRNA expression in bradyzoites from mice. SRS and ApiAP2 genes lacking expression in Groups A-E were excluded. Bz: bradyzoite, Mz: merozoite and CC: cell cycle specific genes. Source data are provided as a Source Data file for this figure.

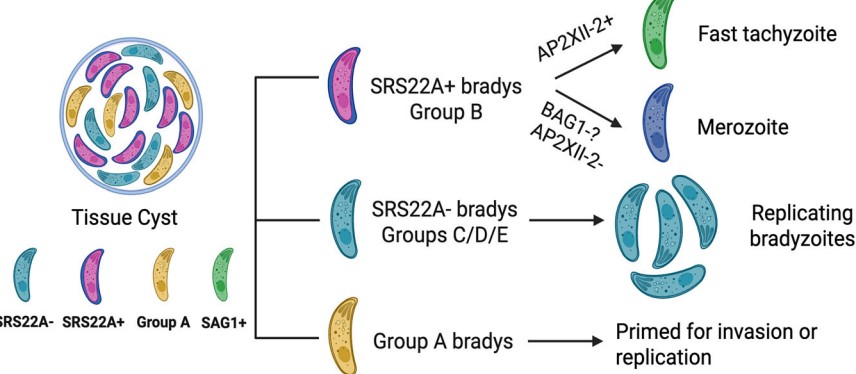

**Fig. 7 | Model illustrating the non-linear life cycle and multifunctional nature of bradyzoites within heterogeneous tissue cysts.** Our findings reveal that brady-zoites both positive and negative for SRS22A antigen can coexist within the same tissue cyst. SRS22A expression distinguishes two bradyzoite subtypes that initiate different developmental pathways: SRS22A+ bradyzoite infection initiates brady-zoite recrudescence into fast-replicating tachyzoites, while infections with SRS22A − bradyzoites initiates bradyzoite-to-bradyzoite replication. A large fraction of SRS22A+ bradyzoites are BAG1-, and it is possible these bradyzoites may initiate the definitive life cycle in feline host, which will require alterations in AP2 factor expression. Rhoptry related gene expression is triggered by egress and invasion events. We have found that a group of in vivo bradyzoites (Group A) possesses this pattern of gene expression, which may help explain the large number of docu-mented parasite invasion events that do not lead to productive infections of host cells. Created in BioRender. Ulu, A. (2026) https://BioRender.com/vup91un.

30 ul aliquot of brain homogenate was deposited on a microscope slide, smeared using a coverslip and allowed to dry. Slides were then fixed in methanol and stored at −20 °C until immunofluorescence staining was performed.

### Detection of parasite burden in mouse tissues

Liver, spleen and lung were harvested from infected CBA mice and stored at −80 °C prior to DNA purification. DNA was purified using a DNeasy minikit (Qiagen) according to manufacturer's instructions. The qPCR reaction used 600 ng of DNA from each organ, 0.375 µM B1 gene primers (Forward:5′- TCCCCTCTGCTGGCGAAAAGT-3′; Reverse: 5′-AGCGTTCGTGGTCAACTATCG-3′), and 2x SensiFAST SYBR No-ROX qPCR master mix (catalog #C755J00, Thomas Scientific, Swedesbro, NJ, USA) in a 20 µl total volume. The qPCR program was: 3 min at 95 °C, 40 cycles of 10 s at 95 °C for denaturation, 30 s at 60 °C for annealing, followed by 5 s at 65 °C and 5 s at 95 °C for extension. The real-time data was collected on a BioRad CFX96 Touch Real-Time PCR Detection system, using Bio-Rad CFX software. A standard curve was generated with serial dilution of purified ME49 genomic DNA and used to cal-culate pg amount from the equation displayed on the linear trendline ($R^2 = 0.99$). All results were expressed as the pg of parasite genomic DNA calculated from an experimentally generated standard curve and relative expression was presented on each graph as compared to naïve controls.

### Western blot analysis of SRS22A

Cysts were purified from the brains of three CBA mice infected with ME49EW for 30 days. Bradyzoites were excysted from the pooled cysts by 90 seconds of pepsin-HCl treatment at 37 °C. RH grown in HFFs were used as a negative control. These parasites were lysed by a 25 G needle and filtered in a 3 µm filter. After washing the parasites, protein lysates were made by adding 100 µL NP-40 buffer with protease inhi-bitors for each $1 \times 10^6$ pelleted parasites. After keeping on ice for 10 minutes, the lysate was centrifuged at 14,000 x $g$ for 10 min at 4 °C, and the supernatant collected. 1 µg of each protein lysate was sepa-rated by SDS-PAGE (15% resolving gel, 5% stacking gel) for 2 h and transferred onto a 0.2 µm nitrocellulose membrane for 35 minutes. Membranes were incubated in 5% milk in 0.1% PBS-T for 30 minutes at room temperature, then probed with rabbit-α-SRS22A antibody (1:500) or pre-immune serum (1:500) overnight at 4 °C. After washing, membranes were probed with goat-α-rabbit HRP conjugate (Bio-Rad #1705046, 1:10,000, Hercules, CA, USA) for one hour at room temperature. The membranes were then reprobed with rabbit-α-PCNA antibody (1:2,500, a gift from Dr. Michael White's laboratory[65,66]) overnight at 4 °C, followed by goat-α-rabbit HRP conjugate for one h at room temperature. Importantly, signal was detected with the Super-Signal™ West Femto Maximum Sensitivity Substrate (ThermoScientific #34094). Membranes were imaged on the Bio-Rad Gel Doc XR+ System at an exposure of 5 seconds.

### Statistical analysis

Statistical analyses for tissue cyst diameter, cyst counts, SRSR22A staining intensity, vacuole counts, and parasite burden were con-ducted in GraphPad Prism (version 10.2.3) using one-way or two-way ANOVA or two-tailed $t$-tests, as appropriate. A $P$-value < 0.05 was considered statistically significant. scRNA-seq data were analyzed using Seurat[67] as described above.

### Reporting summary

Further information on research design is available in the Nature Portfolio Reporting Summary linked to this article.

## Data availability

The scRNA-seq data generated in this study have been deposited in the GEO database under accession code GSE311669. The processed scRNAseq data are also available at GEO. All processed data are also available as Supplementary Data 1 and Supplementary Data 2 in the Supplementary Information. Source data are provided with this paper.

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

## Acknowledgements

*Toxoplasma gondii* gene information was accessed on http://ToxoDB.org. We thank Dr. John Boothroyd for the antibodies used in our study. We would like to acknowledge Mary Hamer, who is the manager of Research Core at UC Riverside School of Medicine for her help with FACS cell sorting operations. This study was supported by NIH NIAID R01 grants AI158417 and AI124682 (M.W.W. and E.H.W.), AI122760 (M.W.W.), and DA048815 (E.H.W) This research used computational resources of the University of California, Riverside HPCC (High-Performance Computing Cluster), which is supported by the NSF MRI-2215705 grant.

## Author contributions

E.H.W. and M.W.W. conceptualized the study. A.U., E.H.W., M.W.W. designed the research, interpreted results and wrote the manuscript. A.U., S.S. and M.W.W. contributed to analysis of the data. A.U., S.S. and N.K. executed all experiments. A.U., E.H.W., M.W.W. and B.L. contributed to analysis and interpretation of the scRNAseq data.

## Competing interests

The authors declare no competing interests.
