## [Transparent Peer Review file · Nature Communications]

Bradyzoite subtypes rule the crossroads of *Toxoplasma* development.

Corresponding Author: Professor Emma Wilson

Version 0:

Reviewer comments:

Reviewer #1

(Remarks to the Author)

This paper provides significant new insights into the biology of *T. gondii*, which is an important human and veterinary pathogen. In particular, the observation on bradyzoite subtypes provides a rigorous definition of the heterogeneity of the *T. gondii* cyst (as established by work by Sinai's laboratory (PMID:26350965 PMCID:PMC4600105)). The identification of SRS22A as a marker for bradyzoites that are committed to the rapid growth (and merozoite transition) in the cat intestine is an important finding for understanding this developmental pathway. Furthermore the identification of SRS9 positive SRS22A negative bradyzoites as a group committed to brady to brady growth is also an important insight into infection dynamics. The ability to FACS sort these populations is an important observation and a significant advance for dissection of what is going on in the tissue cyst. As has been elucidated by other investigators, the in vivo cyst is different/ more mature than the cysts formed in vitro using high pH or low CO₂ differentiation conditions. The scRNA data supports the staining and in vitro data on FACS sorted bradyzoites and establishes the presence of 5 populations (although functionally this may only be 3, however this remains to be determined).

A few questions and suggestions

1. Did the investigators try to examine in vitro long term cultures in muscle myoblasts (PMID: 38213326) or similar cells. The data suggests that cysts formed in these culture systems may be more similar to in vivo cysts? This longer term in vitro culture data (day 21 etc.) might be interesting as a comparison for your scRNA data in vivo. However, given the difficulty of this in vitro myoblast culture system it is not essential for the current manuscript.
2. To that end, it would be useful if it was more clearly stated what the data of in vivo cysts would look like at 2 to 4 days after formation of cysts; as the culture systems are being examined at very early time points after formation of the bradyzoite parasitophorous vacuole
3. Work by Sinai should be added to the paper in the discussion of cyst heterogeneity (e.g. PMID:26350965, PMID: 40492719, PMID: 3673022)
4. As to scRNA there is also work by Sugi on this (as it applies to host cells), e.g. PMID: 35372115. Is there anything in this data that corresponds to what was found in your scRNA For the in vitro cysts.
5. There is no accession number for the scRNA data, as noted this needs to be deposited in GEO and an accession number provided. I also suggest this data be provided to ToxoDB (EuPathDB) for incorporation into that widely used web resource.
6. References 25 and 26 are not fully cited (the titles are cut off).
7. For BAG1/BAG5 you are missing PMID: 10027984 which describes the phenotype of a BAG1 knockout.

Reviewer #2

(Remarks to the Author)

Summary and Overall Assessment

This manuscript presents evidence that native *Toxoplasma gondii* ME49EW cysts harbor heterogeneous bradyzoite subtypes distinguished by SRS22A expression. SRS22A⁺ bradyzoites preferentially recrudescence to fast-growing tachyzoites, whereas SRS22A⁻ bradyzoites tend toward brady-brady replication and cyst-wall reinforcement. Single-cell RNA-seq (~4,190 excysted bradyzoites) resolves five clusters (A–E): Group B is enriched for SRS22A and frequently lacks BAG1, and Group E shows S/M-biased “replicative” signatures. Conventional in-vitro models fail to detect SRS22A protein. The question is important and the dataset is potentially impactful; however, several controls, standardizations, and analyses are required to rigorously support the central claims.

Major Strengths

- Addresses a core gap in native bradyzoite biology using complementary approaches (IFA/FACS, ex vivo astrocyte infection, mouse infection, 10x scRNA-seq).
- Proposes a mechanistic framework linking bradyzoite subtypes to divergent developmental outcomes.
- Generates a resource that could recalibrate how we model chronic infection in vitro.

Suggested Revisions

1. Antibody validation and specificity for SRS22A: You generated a polyclonal antiserum against a single SRS22A peptide and show bradyzoite-specific bands/staining, but there's no peptide competition (antigen pre-absorption) control, isotype control, or demonstration of cross-reactivity against closely related SRS22 paralogs (SRS22i, SRS22E/G) that you discuss extensively. Add peptide competition IFA and immunoblot; test binding to recombinant/overexpressed SRS22 paralogs, or perform CRISPR knockout/conditional knockdown rescue of SRS22A to demonstrate loss and specificity of signal.
2. Reliance on two different ME49 lineages for in vivo vs in vitro. Key contrasts hinge on ME49EW (maintained in vivo) vs ME49B7 (adapted to culture) and RH (Type I) for some blots. Genetic and epigenetic divergence makes it hard to attribute differences (e.g., absence of SRS22A in vitro) to "model limitations" rather than strain history. Please (i) use the same lineage wherever possible (e.g., derive in vitro bradyzoites from ME49EW via limited short-term passage), or (ii) complement with additional Type II lab strains and show the phenotype generalizes. At minimum, acknowledge and temper causal claims.
3. FACS gating and sorting controls. The core conclusion (distinct developmental fates of SRS22A+ vs SRS22A- bradyzoites) rests on antibody-based sorting. Provide full gating strategy, FMO/isotype controls, doublet discrimination, viability dye, compensation matrices, sort purity checks, and a re-sort or post-sort IFA to confirm identity and purity of each fraction. Include replicate sorts across independent mouse infections and show between-experiment variance.
4. Causality vs correlation of SRS22A. Data support that SRS22A+ bradyzoites are associated with tachyzoite recrudescence, but do not prove SRS22A is functionally required. Consider genetic perturbation (SRS22A KO/conditional depletion or overexpression) to test whether altering SRS22A levels changes the brady-tachy vs brady-brady outcome. If not feasible now, the manuscript should clearly reframe the claim as associative and propose SRS22A as a marker, not a driver.
5. Alkaline stress model conclusions. The statement that conventional in vitro models "do not form SRS22A+ bradyzoites" is based on 48-h pH 8.2 HFF culture. Test additional differentiation protocols (longer durations, neuronal/astrocytic cells, hypoxia, different CO₂, serum deprivation, host cytokine milieu) before making general claims. Alternatively, narrow the claim to the specific conditions tested.
6. Interpretation about merozoite priming. The suggestion that Group B bradyzoites may be poised for feline merozoite development is speculative. Either (i) provide orthogonal evidence (e.g., AP2Vlla-1 protein, promoter accessibility, or functional data in feline enterocytes/organoids), or (ii) clearly label as hypothesis and tone down the conclusion.
7. As a suggestion for Figure 7 and the discussion, we encourage you to integrate recent evidence that elements of the chronic-stage program can initiate independently of BFD1 (Robert et al., EMBO Mol. Med., 2025). Concretely, testing BFD1 dependence in your scRNA-seq by examining BSM and BCLA across clusters A-E would help identify BFD1-dependent versus BFD1-independent subpopulations and allow you to incorporate those assignments directly into the model and narrative. We also recommend situating your framework explicitly within a continuum—rather than a binary switch—by comparing your inferred trajectories to proposals of epigenetically primed, transcriptionally poised, context-responsive intermediate states (Shahinas et al., COM, 2025), and mapping clusters A-E onto that spectrum to highlight points of convergence and divergence. Finally, it would be valuable to assess whether your dataset contains a hybrid P6-like population co-expressing tachyzoite and bradyzoite markers (Xue et al., Boothroyd JC. eLife, 2020). Marker overlays, module scores, and cell-cycle-aware analyses could clarify whether Group B or E aligns with this hybrid profile and whether such cells act as progenitor-like intermediates within the bradyzoite lineage.

Reviewer #3

(Remarks to the Author)

This manuscript represents a significant advance in our understanding of *Toxoplasma gondii* bradyzoite biology. The authors provide compelling evidence that bradyzoite populations are heterogeneous and that distinct subgroups possess unique developmental potentials. Their findings explain several long-standing observations related to bradyzoite infection dynamics and reactivation.

The authors identified a novel surface marker, SRS22A, which is specifically expressed on bradyzoites derived from in vivo tissue cysts. They also demonstrated that SRS22A-positive and SRS22A-negative bradyzoites have distinct functions—the former are capable of robust conversion to fast-replicating tachyzoites in astrocytes, while the latter appear to participate primarily in bradyzoite-bradyzoite replication. Supporting this finding, they show using an in vivo infection model that SRS22A-positive bradyzoites disseminate more efficiently and form tissue cysts more readily than their SRS22A-negative counterparts. Finally, using single-cell RNA sequencing, they identified five distinct bradyzoite subpopulations (Groups A-E) within in vivo cysts. Each group exhibits unique transcriptional profiles suggestive of specific developmental trajectories. For

example, Group B bradyzoites—expressing the highest levels of SRS22A—appear transcriptionally primed for conversion to tachyzoites or merozoites, whereas Groups C and D, which lack SRS22A expression, show elevated expression of cyst wall mRNAs consistent with continued bradyzoite replication.

Collectively, these results convincingly demonstrate that bradyzoite development is not a linear process and that multiple developmental fates coexist within chronic-stage cysts. This work provides strong evidence for the functional heterogeneity of bradyzoite populations and offers a refined framework for understanding chronic toxoplasmosis.

The study is well-designed, the experiments are appropriately controlled, and the manuscript is clearly written. The authors also transparently address the limitations of their approach. Importantly, this work opens several exciting avenues for future research, such as identifying the triggers that drive each bradyzoite subgroup's developmental trajectory. In this reviewer's opinion, the quality, novelty, and depth of this study meet the standards expected for Nature Communications.

Minor comment:

Page 8, line 2: The text should refer to Figure 2B rather than 2A.

Version 1:

Reviewer comments:

Reviewer #2

(Remarks to the Author)

The authors have thoroughly and effectively addressed all of my prior comments. The revised manuscript is substantially strengthened by clearer methodological descriptions, improved figures, and the addition of key control data and analyses, including full FACS gating/purity documentation and trajectory analysis. The claims regarding SRS22A have been appropriately refined, antibody specificity is convincingly justified, and interpretations of in-vitro models and developmental trajectories are now well balanced. Speculative elements have been clearly framed as hypotheses. The manuscript is now rigorous and coherent and, in my opinion, represents an interesting and valuable study that provides the community with an important new resource for understanding chronic stage(s) of *Toxoplasma gondii*.

Response to Reviewer Comments

We thank all the reviewers for their time and comments on our manuscript. We answered their questions point-by-point below in bold font, and modified the manuscript accordingly. The modifications were highlighted in the manuscript in red in the Marked copy.

Reviewer's Comments:

Reviewer #1 (Remarks to the Author)

This paper provides significant new insights into the biology of *T. gondii*, which is an important human and veterinary pathogen. In particular, the observation on bradyzoite subtypes provides a rigorous definition of the heterogeneity of the *T. gondii* cyst (as established by work by Sinai's laboratory (PMID:26350965 PMID:PMC4600105)). The identification of SRS22A as a marker for bradyzoites that are committed to the rapid growth (and merozoite transition) in the cat intestine is an important finding for understanding this developmental pathway. Furthermore the identification of SRS9 positive SRS22A negative bradyzoites as a group committed to brady to brady growth is also an important insight into infection dynamics. The ability to FACS sort these populations is an important observation and a significant advance for dissection of what is going on in the tissue cyst. As has been elucidated by other investigators, the *in vivo* cyst is different/ more mature than the cysts formed *in vitro* using high pH or low CO₂ differentiation conditions. The scRNA data supports the staining and *in vitro* data on FACS sorted bradyzoites and establishes the presence of 5 populations (although functionally this may only be 3, however this remains to be determined).

We appreciate the reviewer's assessment and recognition of the significance of the work.

A few questions and suggestions

1. Did the investigators try to examine *in vitro* long term cultures in muscle myoblasts (PMID: 38213326) or similar cells. The data suggests that cysts formed in these culture systems may be more similar to *in vivo* cysts? This longer term *in vitro* culture data (day 21 etc.) might be interesting as a comparison for your scRNA data *in vivo*. However, given the difficulty of this *in vitro* myoblast culture system it is not essential for the current manuscript.

We thank the reviewer for this suggestion and agree the myoblast culture system offers one of the best options for long term culture and *in vitro* propagation of cysts. Such culture systems will likely be valuable to directly assess questions of cyst maturity *in vitro*. Of note, the Luder lab examined parasite gene expression in four

different murine cell types including neurons, astrocytes, fibroblasts and skeletal muscle cells¹. The expression of SRS22A was very low in all four cell types agreeing with our findings with the alkaline-stress model and *ex vivo* recrudescence models shown in Figure 2. We have now included this citation in the paper on page 15 (in red). A more complete evaluation of the available *in vitro* bradyzoite developmental models will be an important future addition to the field.

2. To that end, it would be useful if it was more clearly stated what the data of *in vivo* cysts would look like at 2 to 4 days after formation of cysts; as the culture systems are being examined at very early time points after formation of the bradyzoite parasitophorous vacuole.

The question of when functional bradyzoites are present in mice was historically examined in an early 1997 paper² using *in vivo* bradyzoites to orally infect naive mice. It takes about a week for functional tissue cysts to appear in murine infections. In our 2023 mBio paper we show that bradyzoites first differentiate into fast-growing tachyzoites and only reform bradyzoites after a shift to slower growth at Day 6-7, which explains why it takes a week in animals. The earliest cysts we examined (14dpi, Figure 1) will therefore be only 7 days after their formation. In addition, detailed kinetics of cyst dynamics by the Sinai group analyzed growth in cysts from 21dpi to 56dpi with overall average size increasing over time. Our analysis of cysts even earlier than this first time point with an average cyst size <20um (smaller than some of our *in vitro* cysts, Figure 3B) suggests we are indeed analyzing relatively recently formed cysts. We have clarified and expanded on these points in the Figure 1 experimental methods (see page 19 Methods, in red).

3. Work by Sinai should be added to the paper in the discussion of cyst heterogeneity (e.g. PMID:26350965, PMID: 40492719, PMID: 3673022)

As requested, we have included these citations on pages 5 (Results), 15 (Discussion).

4. As to scRNA there is also work by Sugi on this (as it applies to host cells), e.g. PMID: 35372115. Is there anything in this data that corresponds to what was found in your scRNA For the *in vitro* cysts.

We thank the reviewer for bringing this to our attention. The paper by Sugi et al revealed several clusters of host cell populations following infection and *in vitro* differentiation to bradyzoites supporting the concept of parasite heterogeneity at the bradyzoite level. In that work, the clustering reflects heterogeneity in the *host cell transcriptomes* of bradyzoite-infected HFFs rather than in the parasites themselves. In contrast, our analysis focuses on the transcriptional diversity among *bradyzoites* within cysts, distinguishing our findings from those reported by Sugi et al. We incorporated this point with a statement in the Discussion (page 16) as below:

“...Another study found that BAG1⁺ bradyzoite-infected HFFs form two distinct transcriptional subsets, suggesting that variability in host responses may reflect heterogeneity among the bradyzoites³.”

5. There is no accession number for the scRNA data, as noted this needs to be deposited in GEO and an accession number provided. I also suggest this data be provided to ToxoDB (EuPathDB) for incorporation into that widely used web resource.

We deposited our data to GEO (Accession number: GSE311669), and will release publishing upon acceptance. In addition, we would be delighted to deposit our data with ToxoDb upon invitation as soon as it is accepted so that it is easily accessible by Toxoplasma research community.

6. References 25 and 26 are not fully cited (the titles are cut off).

We corrected those references as suggested.

7. For BAG1/BAG5 you are missing PMID: 10027984 which describes the phenotype of a BAG1 knockout.

We thank the reviewer for pointing out the missing citation on BAG1 knockout strains. We now cite all BAG1 knockout references in the paper.

Reviewer #2 (Remarks to the Author)

Summary and Overall Assessment

This manuscript presents evidence that native *Toxoplasma gondii* ME49EW cysts harbor heterogeneous bradyzoite subtypes distinguished by SRS22A expression. SRS22A⁺ bradyzoites preferentially recrudescence to fast-growing tachyzoites, whereas SRS22A⁻ bradyzoites tend toward brady-brady replication and cyst-wall reinforcement. Single-cell RNA-seq (~4,190 excysted bradyzoites) resolves five clusters (A–E): Group B is enriched for SRS22A and frequently lacks BAG1, and Group E shows S/M-biased “replicative” signatures. Conventional in-vitro models fail to detect SRS22A protein. The question is important and the dataset is potentially impactful; however, several controls, standardizations, and analyses are required to rigorously support the central claims.

Major Strengths

- Addresses a core gap in native bradyzoite biology using complementary approaches (IFA/FACS, ex vivo astrocyte infection, mouse infection, 10x scRNA-seq).
- Proposes a mechanistic framework linking bradyzoite subtypes to divergent developmental outcomes.
- Generates a resource that could recalibrate how we model chronic infection in vitro.

We thank the reviewer for recognizing the data as potentially impactful and important.

Suggested Revisions

1. Antibody validation and specificity for SRS22A: You generated a polyclonal antiserum against a single SRS22A peptide and show bradyzoite-specific bands/staining, but there's no peptide competition (antigen pre-absorption) control, isotype control, or demonstration of cross-reactivity against closely related SRS22 paralogs (SRS22i, SRS22E/G) that you discuss extensively. Add peptide competition IFA and immunoblot; test binding to recombinant/overexpressed SRS22 paralogs, or perform CRISPR knockout/conditional knockdown rescue of SRS22A to demonstrate loss and specificity of signal.

The SRS antigens are a related gene family, however, they are also a divergent set of proteins. Overall, SRS protein sequences share only ~40% identity that is confined to specific segments of protein structure, while other regions of SRS protein sequence show little to no conservation. This feature has permitted successful antibody production in long term use for specific SRS proteins (e.g. commonly used SAG1 and SRS9 antibodies). To raise antibodies against SRS22A, we selected a peptide (20 amino acids) representing SRS22A protein sequence that is unique among SRS proteins as well as all other proteins encoded in the *Toxoplasma* genome. For example, the selected SRS22A peptide sequence shares one amino acid with SRS22G protein sequence (see example alignment of SRS22A vs SRS22G, 22i and 22E below, peptide sequence highlighted). For SRS22i only four noncontiguous amino acids are shared with SRS22A. Consequently, there is very little chance the anti-SRS22A antisera raised against this peptide will cross react with other SRS antigens, which is borne out by the unique expression patterns of SRS22A in bradyzoites and tachyzoite infections. Moreover, SRS22A mRNA is expressed at levels 20-75 fold higher in our bradyzoite subtypes than either SRS22i or SRS22G mRNAs decreasing the near zero chance of cross reactivity to these antigens to even less.

Query: SRS22A Query ID: lcl|Query_2679595 Length: 194

>SRS22G

Sequence ID: Query_2679597 Length: 176
Range 1: 1 to 176

Score:142 bits(358), Expect:4e-48,
Method:Compositional matrix adjust.,
Identities:81/195(42%), Positives:116/195(59%), Gaps:20/195(10%)

SRS22A	1	MKFSLLTLGALAIQAQASAL	RGNDGRSSRVIEKEAEVAK	VC-LTTPLSFDIAEAGQSV	59
		MKFSLLTLGALA SA QA+ ++G++ + +		NVC + L+F++ AG+SV	
SRS22G	1	MKFSLLTLGALAFSAHQAAIVQGEETTQ	-----	PKPNNVCSANSSTLTFNLTRAGESV	54
SRS22A	60	TFKCDKTLKYLDPALDSKNPKMYKGSNPVTIHDFLPSASL	TEVTAADDQVSETQRGSNG	119	
		F C +T+ LDP ++ P+MY+G++ V I +FLP A L EV + D + R S			
SRS22G	55	VFTCGETVTTLDPEFNATFPEMYEGNSKVRILEFLPDAKLEEVKNSVD	-----	SSRASVS	109
SRS22A	120	SDSEKEKEYKFTVPTLPSEQDLHVYCKAVDPSTAERTENPNACQVTFHIASSAVRPFLA	179		
		++ Y FTVP LPS++ LHV C A + + CQV FHIASSA+RP +A			
SRS22G	110	GNT-----YNFTVPVLPSEHHLHVNCTAAGEKSKSTVTD	---	CQVFFHIASSAIRPVMA	161
SRS22A	180	AGVVAGVIASVLQFA	194		
		A + ++AS+LQFA			
SRS22G	162	ASAIVSLVASLLQFA	176		

Query: SRS22A Query ID: lcl|Query_6353935 Length: 194

>SRS22i

Sequence ID: Query_6353937 Length: 179
Range 1: 1 to 179

Score:152 bits(383), Expect:8e-52,
Method:Compositional matrix adjust.,
Identities:89/194(46%), Positives:115/194(59%), Gaps:15/194(7%)

SRS22A	1	MKFSLLTLGALAIQAQASAL	RGNDGRSSRVIEKEAEVAK	NVCLTTPLSFDIAEAGQSVT	60
		MKFSLLTLGALA SAQQAS +RG +S K+ N + LSF+I AG+SV			
SRS22i	1	MKFSLLTLGALAFSAQASLVRGEGDNTSGQOAKDGVCPN	---	SSLSFNITRAGESVL	56
SRS22A	61	FKCDKTLKYLDPALDSKNPKMYKGSNPVTIHDFLPSASL	TEVTAADDQVSETQRGSNGS	120	
		F C +T+ LDPA P+MY+G++ V I +FLP A+L +V + + S+GS			
SRS22i	57	FTCSETVSNLDPAFSDTIPEMYEGNSKVRILEFLPGATLQKVETTGESR	-----	SSGS	109
SRS22A	121	DSEKEKEYKFTVPTLPSEQDLHVYCKAVDPSTAERTENPNACQVTFHIASSAVRPFLAA	180		
		K + FTVP LPS++ LHV C ST + C+VTF+IASSAVR LA			
SRS22i	110	G-----KTFNFTVPILPSDEHQLHVNCTKTASTFRNGDENKDCKVTFNIASSAVRAGLAM	165		
SRS22A	181	GVVAGVIASVLQFA	194		
		V GV+AS+LQFA			
SRS22i	166	SAVVGVASLLQFA	179		

Query: SRS22A Query ID: lcl|Query_5481433 Length: 194

>SRS22E

Sequence ID: Query_5481435 Length: 181
Range 1: 1 to 181

Score:116 bits(290), Expect:8e-38,
Method:Compositional matrix adjust.,
Identities:88/194(45%), Positives:119/194(61%), Gaps:13/194(6%)

```
SRS22A 1 MKFSLLTLGALAIQAQASALRGNDGRSSRVIEKEAEVAKNVCLTTPLSFDIAEAGQSVT 60
MKFSLLTLGALA++AQQ SA+ ++ S K+ +KN LSF+I +AG+ +
SRS22E 1 MKFSLLTLGALALAAQQTSAVAADNTLESAQQAQKDTTCSKN-----KSLSFNITQAGEFIL 56

SRS22A 61 FKCDKTLKYLDPALDSKNPKMYKGSNPVTIHDFLPSASLTVTAADDQVSETQRGSNGS 120
F C + + LDPA ++ P+M++G N V I DFLP A+L VT A + +
SRS22E 57 FTCGEDVPTLDPAFNASTPEMFEQDNRVKIRDFLPGATLENTTADAATATLVASATAA- 115

SRS22A 121 DSEKEKEYKFTVPTLPSEQQLHVVYCKAVDPSTAERTENPNACQVTFHIIASSAVRPFLAA 180
+Y FTVP+LPS+ LHVYC+ ++ R E N C+VTFHIIASSAVRP +AA
SRS22E 116 -----KYNFTVPSLPSDDHSLHVYCR--KDASKTREEEKNECKVTFHIIASSAVRPVMAA 167

SRS22A 181 GVVAGVIASVLQFA 194
VAGV+AS+L FA
SRS22E 168 SAVAGVVASLLHFA 181
```

2. Reliance on two different ME49 lineages for in vivo vs in vitro. Key contrasts hinge on ME49EW (maintained in vivo) vs ME49B7 (adapted to culture) and RH (Type I) for some blots. Genetic and epigenetic divergence makes it hard to attribute differences (e.g., absence of SRS22A in vitro) to “model limitations” rather than strain history. Please (i) use the same lineage wherever possible (e.g., derive in vitro bradyzoites from ME49EW via limited short-term passage), or (ii) complement with additional Type II lab strains and show the phenotype generalizes. At minimum, acknowledge and temper causal claims.

The ME49EW strain was chosen because it is a Type II strain that has not been adapted to cell culture as this causes major permanent developmental changes. ME49B7 is used here as a stable HFF-adapted strain to measure SRS22A expression in the most commonly used method for *in vitro* bradyzoite induction (alkaline-stress). This was compared to spontaneous induction of bradyzoites in the *ex vivo* recrudescence model we developed for the ME49EW strain. There was no SRS22A expression in either *in vitro* model indicating this is not due to strain differences. The use of RH tachyzoites is a negative control in Western Blots and was chosen because it is unable to differentiate. We have clarified in the paper that SRS22A was not expressed in either *in vitro* model we tested (Abstract, and page 15, in red). Future work will be needed to fully evaluate other *in vitro* models as to the ability of these models to form the different bradyzoite subtypes, which is beyond the scope of this paper.

3. FACS gating and sorting controls. The core conclusion (distinct developmental fates of SRS22A+ vs SRS22A- bradyzoites) rests on antibody-based sorting. Provide full gating strategy, FMO/isotype controls, doublet discrimination, viability dye, compensation matrices, sort purity checks, and a re-sort or post-sort IFA to confirm identity and purity of each fraction. Include replicate sorts across independent mouse infections and show between-experiment variance.

We apologize that we did not provide all control data for the sorting analysis and thank the reviewer for giving us the opportunity to include the following data, all of which are now included in the Supplemental files, Supplemental Fig. S2A-D.

- 1) Whole gating strategy with doublet discrimination.
- 2) Pre- and post-sort viability and purity data of bradyzoite populations.
- 3) Reproducibility of FACS sorting in experiments run on 4-5 different days.
- 4) With regards to FMO control and compensation matrices, since we only have one antibody, we only included an unstained control FACS plot.

This additional information is included in Methods, Section 'FACS sorting of bradyzoite populations'

“Full gating strategy, controls and reproducibility data were shown in Fig. S2A-D. FACS purification of the SRS22A+ and SRS9+ populations was highly reproducible as evidenced by low variability obtained from separate experiments run on different dates (Fig. S2B-D).”

We included below the new Fig. S2 for ease of access to the reviewer:

A. SRS22A FACS Control Plots and Purity of the Populations

B. SRS22A Full Gating Strategy

SRS22A FACS Reproducibility and unstained Control

Viability of Day 0 Bradyzoites and Purity of the FACS-sorted Populations

4. Causality vs correlation of SRS22A. Data support that SRS22A+ bradyzoites are associated with tachyzoite recrudescence, but do not prove SRS22A is functionally required. Consider genetic perturbation (SRS22A KO/conditional depletion or overexpression) to test whether altering SRS22A levels changes the brady-tachy vs brady-

brady outcome. If not feasible now, the manuscript should clearly reframe the claim as associative and propose SRS22A as a marker, not a driver.

At no point in the paper did we propose that SRS22A was responsible for driving bradyzoite development, indeed our goal was, as stated on page 5, “to identify potential *in vivo* bradyzoite markers”. We apologize if mechanism was implied. The function of SRS antigens has been a long and torturous goal that has largely been fruitless. SRS22A is used in this study as marker of bradyzoite development. Other findings in this study do suggest avenues for future investigation in particular the differential regulation of ApiAP2 transcription factors.

5. Alkaline stress model conclusions. The statement that conventional *in vitro* models “do not form SRS22A+ bradyzoites” is based on 48-h pH 8.2 HFF culture. Test additional differentiation protocols (longer durations, neuronal/astrocytic cells, hypoxia, different CO₂, serum deprivation, host cytokine milieu) before making general claims. Alternatively, narrow the claim to the specific conditions tested.

An exhaustive comparison of *in vitro* models is not the focus of this research and as discussed above we have modified the text to clarify this and the claims that we are making (see above and text in red on pages 5, 6, 15).

6. Interpretation about merozoite priming. The suggestion that Group B bradyzoites may be poised for feline merozoite development is speculative. Either (i) provide orthogonal evidence (e.g., AP2Vlla-1 protein, promoter accessibility, or functional data in feline enterocytes/organoids), or (ii) clearly label as hypothesis and tone down the conclusion.

As part of the original discussion we wrote the sentence below that demonstrates we are in complete agreement with the reviewer that we do not currently know what drives tachyzoite to merozoite development (page 12).

“Altogether, these features may indicate Group B bradyzoites are poised to initiate merozoite development in the cat. Further study will be required to understand whether Group B bradyzoites or another *in vivo* bradyzoite group are the primary drivers of bradyzoite-to-merozoite development.”

7. As a suggestion for Figure 7 and the discussion, we encourage you to integrate recent evidence that elements of the chronic-stage program can initiate independently of BFD1 (Robert et al., EMBO Mol. Med., 2025). Concretely, testing BFD1 dependence in your scRNA-seq by examining BSM and BCLA across clusters A–E would help identify BFD1-dependent versus BFD1-independent subpopulations and allow you to incorporate those assignments directly into the model and narrative. We also recommend situating your framework explicitly within a continuum—rather than a binary switch—by comparing your inferred trajectories to proposals of epigenetically primed, transcriptionally poised, context-responsive intermediate states (Shahinas et al., COM, 2025), and mapping

clusters A–E onto that spectrum to highlight points of convergence and divergence. Finally, it would be valuable to assess whether your dataset contains a hybrid P6-like population co-expressing tachyzoite and bradyzoite markers (Xue et al., Boothroyd JC. eLife, 2020). Marker overlays, module scores, and cell-cycle-aware analyses could clarify whether Group B or E aligns with this hybrid profile and whether such cells act as progenitor-like intermediates within the bradyzoite lineage.

To aid interpretation and incorporation of previously proposed bradyzoite differentiation pathways we examined BFD1, BSM and BCLA. BFD1 is lowly expressed across all clusters, and thus may not be applicable to identify BFD1-dependent/independent populations. In contrast BSM is highly expressed across all clusters again minimizing its use for identifying parasite trajectories. BCLA, associated with cysts, is the most differentially expressed being somewhat excluded from Group B. The UMAP projections for all these genes are provided here for the reviewer and in a new Supplemental Figure, Fig. S5A. We also discuss this analysis on page 18 (see text in red). Below are the violin plots and UMAPs associated with BFD1.

Supplementary Figure S5

Similarly, genes mentioned in the P6 cluster (which defined a cluster of parasites separated from the alkaline-induced population) in the paper Xue et al., Boothroyd JC. eLife, 2020 are mostly lowly expressed or not defined to a single cluster. We included the corresponding UMAPs below. While some genes appear to map onto Group B others are minimally expressed or not defined to a single cluster. Therefore, we did not include these additional analyses but provide all data via its deposit in GEO (Accession number: GSE311669).

To assist with interpretation and analysis of the scRNAseq data set, a trajectory analysis is now provided (Fig. S5B). This supports one of our main conclusions that bradyzoite heterogeneity is a developmental process that defines their function by identifying at least one branching point. This analysis was performed by Dr. Brandon Le, who is the Bioinformatics Core Coordinator at UC Riverside Institute for Integrative Genome Biology (IIGB). Due to his significant contribution, we now include him as a co-author in our manuscript.

Interpretation of this additional analysis is on page 14 and Fig. S5B:

“Finally, trajectory analysis, although not definitive, supports the presence of alternative bradyzoite developmental programs as seen with SRS22A+ and SRS22A- parasites. The inferred trajectory begins with parasites in Group D, progresses toward the transitional Group C/D population, and then bifurcates towards either Group B (SRS22A+) or Group A (Fig. S5B).”

Model: slingshot

Distinct trajectory plots using the model: slingshot

Finally, we also mapped genes involved in the epigenetic regulation of stage transition as requested by the reviewer. However, none of these genes passed the p-value cutoff or displayed distinguishing patterns across our clusters, and therefore we could not draw any conclusive inferences from this analysis. As a result, this analysis did not clarify the potential roles of the MORC/HDAC3 repressor complex or the AP2 factors.

In the paper by Shahinas et al⁴, we emphasized the importance of MORC/HDAC3 complex in chromatin remodeling and stage transition. We similarly examined genes implicated in merozoite stage transition, including AP2XII-1 and AP2X1-2, whose combined deletion is known to initiate sexual stage transition *in vitro*. Yet, none of these genes met our statistical thresholds. For completeness, we provide the UMAPs for these genes below.

To respond to this question, we now included those citations and a statement on page 17:

“... Turning off BAG1 expression may signal a priming event that pre-stages SRS22A+ Group B bradyzoites to recrudesce into the tachyzoite or merozoite stage depending on the host environment. Consistent with this, epigenetic priming events were proposed to be responsible for stage transition. For example, one hypothesis involves regulation of chromatin remodeling by the MORC/HDAC3 repressor complex^{4, 5}.”

Reviewer #3 (Remarks to the Author):

This manuscript represents a significant advance in our understanding of *Toxoplasma gondii* bradyzoite biology. The authors provide compelling evidence that bradyzoite populations are heterogeneous and that distinct subgroups possess unique developmental potentials. Their findings explain several long-standing observations related to bradyzoite infection dynamics and reactivation.

The authors identified a novel surface marker, SRS22A, which is specifically expressed on bradyzoites derived from in vivo tissue cysts. They also demonstrated that SRS22A-positive and SRS22A-negative bradyzoites have distinct functions—the former are capable of robust conversion to fast-replicating tachyzoites in astrocytes, while the latter appear to participate primarily in bradyzoite-bradyzoite replication. Supporting this finding, they show using an in vivo infection model that SRS22A-positive bradyzoites disseminate more efficiently and form tissue cysts more readily than their SRS22A-negative counterparts.

Finally, using single-cell RNA sequencing, they identified five distinct bradyzoite subpopulations (Groups A–E) within in vivo cysts. Each group exhibits unique transcriptional profiles suggestive of specific developmental trajectories. For example, Group B bradyzoites—expressing the highest levels of SRS22A—appear transcriptionally primed for conversion to tachyzoites or merozoites, whereas Groups C and D, which lack SRS22A expression, show elevated expression of cyst wall mRNAs consistent with continued bradyzoite replication.

Collectively, these results convincingly demonstrate that bradyzoite development is not a linear process and that multiple developmental fates coexist within chronic-stage cysts. This work provides strong evidence for the functional heterogeneity of bradyzoite populations and offers a refined framework for understanding chronic toxoplasmosis.

The study is well-designed, the experiments are appropriately controlled, and the manuscript is clearly written. The authors also transparently address the limitations of their approach. Importantly, this work opens several exciting avenues for future research, such as identifying the triggers that drive each bradyzoite subgroup's developmental trajectory.

In this reviewer's opinion, the quality, novelty, and depth of this study meet the standards expected for Nature Communications.

We thank the reviewer for their positive feedback and thoughtful analysis of our manuscript, as well as for recognizing its significance.

Minor comment:

Page 8, line 2: The text should refer to Figure 2B rather than 2A.

We thank the reviewer for catching this mistake, we now corrected this as suggested.

References*

***References below have different numbers in the main manuscript and solely included here for easy access to the reviewers.**

1. Swierzy, I.J. *et al.* Divergent co-transcriptomes of different host cells infected with *Toxoplasma gondii* reveal cell type-specific host-parasite interactions. *Sci Rep* **7**, 7229 (2017).
2. Dubey, J.P. Bradyzoite-induced murine toxoplasmosis: stage conversion, pathogenesis, and tissue cyst formation in mice fed bradyzoites of different strains of *Toxoplasma gondii*. *J Eukaryot Microbiol* **44**, 592-602 (1997).
3. Sugi, T. *et al.* Single Cell Transcriptomes of In Vitro Bradyzoite Infected Cells Reveals *Toxoplasma gondii* Stage Dependent Host Cell Alterations. *Front Cell Infect Microbiol* **12**, 848693 (2022).

4. Shahinas, M., Pachano, B., Robert, M.G., Swale, C. & Hakimi, M.A. Decoding the epigenetic blueprint behind *Toxoplasma* (pre)sexual commitment and chronic persistence. *Curr Opin Microbiol* **88**, 102662 (2025).
5. Xue, Y. *et al.* A single-parasite transcriptional atlas of *Toxoplasma Gondii* reveals novel control of antigen expression. *Elife* **9** (2020).